# Single rosette-based generation of uniform cortical assembloids recapitulating cellular interactions between neurons and glial cells

Eunjee Kim[1,5], Yunhee Kim [1,2,5], Soojung Hong[1,2], Inha Kim [1,2], Juhee Lee [1], Jong-Yeon Yoo[3], Jihyun Kim[4], Kwangmin Yoo[4], Hojin Lee[4], Joung-Hun Kim [3], Jungmin Choi [4] & Kunyoo Shin [1,2] ✉

Despite recent advances, current brain organoid technologies face ongoing challenges in managing heterogeneity and representing the diverse structure and cell types of the human brain. Here, we develop a module-based cellular reconstitution technology to sequentially build uniform cortical assembloids with mature cortical structures and functional connectivity. The uniformity and maturity of the cortical assembloids are achieved by creating single-rosette-based organoids at the early stage, whose sizes were big and consistent with the treatment of Wnt and Hedgehog agonists, followed by spatial reconstitution with the Reelin-expressing neuronal layer and non-neuronal glial cells. The resulting single-rosette-based cortical assembloids are highly uniform and reproducible without significant batch effects, solving major heterogeneity issues caused by difficulties in controlling the number and size of rosettes in conventional multi-rosette organoids. Furthermore, these cortical assembloids structurally and functionally recapitulate the physiology of the human brain, including the six-layered cortical structure, functional connectivity, and dynamic cellular interplay between neurons and glial cells. Our study thus provides an innovative preclinical model to study a range of neurological disorders, understanding the pathogenesis of which requires an organoid system capable of representing the dynamic cellular interactions and the maturity of the human brain.

Brain organoids are three-dimensional (3D), self-organizing tissues derived from pluripotent stem cells that have been developed for several years, starting from whole-brain organoids[1,2] to region-specific organoids—such as cortical organoids[3–5], midbrain organoids[3,6,7], and subpallial organoids[4,8,9]—to connected hybrid organoids[4,10,11]. Unlike conventional neural cell cultures and genetically-modified mouse models, which are not likely to represent the complex dynamics of the cellular interactions present in the human brain, brain organoids mimic many aspects of the tissue architecture and cellular composition of the human brain in vivo, thus serving as an innovative platform for studying the pathophysiology of the human brain[12–14].

Although brain organoids provide a valuable system for studying the human brain, current brain organoids have several critical limitations. First, the heterogeneity in structural and functional maturity between individual organoids and batches is problematic. The reproducibility issue in current brain organoid technologies hinders

[1]Institute of Molecular Biology and Genetics, Seoul National University, Seoul, Republic of Korea. [2]School of Biological Sciences, College of Natural Sciences, Seoul National University, Seoul, Republic of Korea. [3]Department of Life Sciences, Pohang University of Science and Technology, Pohang, Gyeongbuk, Republic of Korea. [4]Department of Biomedical Sciences, Korea University College of Medicine, Seoul, Republic of Korea. [5]These authors contributed equally: Eunjee Kim, Yunhee Kim. ✉e-mail: kunyoos@snu.ac.kr

the rapid advance of this field, mainly due to the challenge of precisely controlling the quality of organoid growth during extended culture periods, which can last up to one year. Particularly, there is significant difficulty in the early control of the number and the size of vesicle-like, neuroepithelial structures called rosettes in conventional multi-rosette organoids. These structures are different from the secondary vesicle of telencephalon[15] in vivo developing human brain, which contains a single ventricular space and becomes forebrain[16]. Additionally, there is a lack of spatial and temporal control of reelin (RELN), which is expressed in the first layer of neurons, migrating into the marginal zone at the early stage of human corticogenesis, and is required for proper neural guidance and laminar organization[17,18]. Although several approaches have been used to increase the maturity of brain organoids, such as employing long-term culture methods[19,20], these organoids still exhibit inconsistency and heterogeneity in their structural and functional maturity due to the lack of quality controls on organoid growth during extended culture periods. Second, despite several methodologies of co-culturing with glial cells, these models still exhibit immature cellular interactions between neurons and glial cells with a high level of inconsistency[21,22]. These limitations further hinder the study of dynamic cellular interactions in the human brain, rendering them incapable of examining cell-type-specific phenotypes and accurately modeling various disease-specific human brains. Therefore, combined with inconsistent development of structural and functional maturity and a low level of cellular diversity, current brain organoid technologies face substantial challenges that impede their ability to practically and reliably model the complexity of the human brain.

In this study, we developed a step-wise, module-based cellular reconstitution technology to sequentially build human cortical assembloids[23]. These single-rosette-based cortical assembloids, developed by sequential reconstitution with non-neuronal glial cells, showed a great degree of consistency and uniformity with minimal heterogeneity and batch effects. Also, these cortical assembloids represented the six-layered cortical structure with mature laminar organization and functional connectivity between neurons and glial cells.

## Results

### Creation of cortical assembloids by a module-based, cellular reconstitution of multiple cell types in human brains

Various approaches have been developed to generate brain organoids that recapitulate several aspects of human brain development. While current cortical organoid technologies have made notable progress in enhancing cellular complexity and reproducibility[1–5,24–26], some areas remain open for further improvement. For example, although multiple studies have reported the presence of glial cells such as astrocytes and microglia, these cells often display uneven spatial distribution within organoids, and their proportions relative to the total cell population can vary across individual organoids. Additionally, the maturation of glial cells, particularly astrocytes, typically requires prolonged culture durations of up to a year, posing practical challenges for consistency and scalability. Furthermore, although the presence of six cortical layers has been described in some studies, including RELN-expressing first-layer neurons[3,20], the complete spatial organization of these layers in the correct sequential order remains incompletely established[2,25,26]. Variability in cell-type composition, structural organization, and functional maturity also persists across batches, protocols, and human pluripotent stem cell (hPSC) lines, as well as within individual organoids. Together, these considerations suggest that further refinement of organoid technologies may help facilitate the development of more mature and consistent cortical organoid models.

To overcome these limitations, we developed a stepwise strategy to create mature cortical assembloids[23], aimed at providing essential developmental cues and reconstituting diverse different cell types within the system (Fig. 1a). Initially, embryoid bodies (EBs) were developed from hPSCs through suspension culture. These EBs were cultured in Matrigel for 7 days for neuroectoderm lineage specification. On day 14, Matrigel was removed, and the neuroectoderm structures were cultured under shaking conditions for additional 11 days, resulting in the formation of forebrain organoids with multiple individual neuroepithelium-like structures, known as rosettes, consisting of neural progenitor cells (NPCs). From days 25 to 32, the Hedgehog (Hh) and Wnt pathways[27,28] were pharmacologically activated to promote the proliferation and expansion of NPCs, as evidenced by an increase of SOX2-positive cells, thickening of the ventricular zone (VZ), and enlargement of each rosette (Fig. 1b, c and Supplementary Fig. 1a), while maintaining dorsal identity (Supplementary Fig. 1b, c).

On day 32, early forebrain organoids with larger rosettes were manually dissected into single-rosette structures to better mimic the single VZ layer of the developing brain (Fig. 1d and Supplementary Fig. 1d). The resulting single-rosette organoids were uniform and cyst-like structures without considerable variation between different lines (Fig. 1d, e and Supplementary Fig. 1d). In all single rosette organoids, single lumens were observed while maintaining the correct apical polarity crucial for proper neuronal development (Fig. 1d and Supplementary Fig. 1e). On day 35, following an additional 3 days of culture to stabilize the dissected structure, five single-rosette organoids whose diameters were closest to the median value were selected and reconstituted using hPSC-derived neurons engineered to express RELN (Fig. 1f, g and Supplementary Fig. 1f). These RELN-expressing neurons exhibited characteristics of glutamatergic neurons, positive for MAP2, VGLUT1, and P73, in addition to the exogenous expression of RELN, with high level of homogeneity (Fig. 1f). Based on the marker expression, the engineered RELN-expressing neurons are likely to function as Cajal-Retzius neurons, which, during human brain development, migrate into the marginal zone (later becoming layer 1) at the early stage of corticogenesis[17,18]. To provide the necessary cues for precise neural guidance conducive to the formation of structured cortical layers, individual single-rosette organoids were encapsulated with a thin Matrigel layer infused with RELN-expressing neurons. The resulting structures were then subjected to an additional culture period of 15 days.

On day 50, glial cell integration into the resulting organoids— cortical assembloids —was achieved via microinjection of hPSCs-derived astrocytes and microglia into the outer cortical layer (Fig. 1h–k and Supplementary Fig. 1g)[29,30]. This step aimed to incorporate essential glial cells, contributing to structural and functional maturation. Subsequently, these assembloids were cultured for an additional 30 days to further promote maturation, such as neuronal connection refinement, glial cell maturation, and functional network establishment, resulting in the development of highly uniform human cortical assembloids closely resembling the complexity and organization of the developing human brain with enhanced cellular diversity and minimized batch effects (Fig. 1l).

### Cortical assembloids mimic the six-layered cortical structure of the human brain

Next, we examined the structural and functional maturity of the cortical assembloids. The neurons in the cortical assembloids exhibited mature characteristics with pyramidal morphology, as demonstrated by complex dendritic structures and neuronal projections, and displayed spontaneous action potentials and robust neuronal activity at the single-cell level (Supplementary Fig. 2a-e). Moreover, these neurons displayed a radially-aligned organization, with axonal projections extending across cortical layers, spanning from the outermost part to the intracortical region of the cortical assembloids (Fig. 2a; Supplementary Fig. 2a, b and Supplementary Movie 1).

To further evaluate the organization of these neurons into layered structures akin to the human cortex, we conducted immunostaining

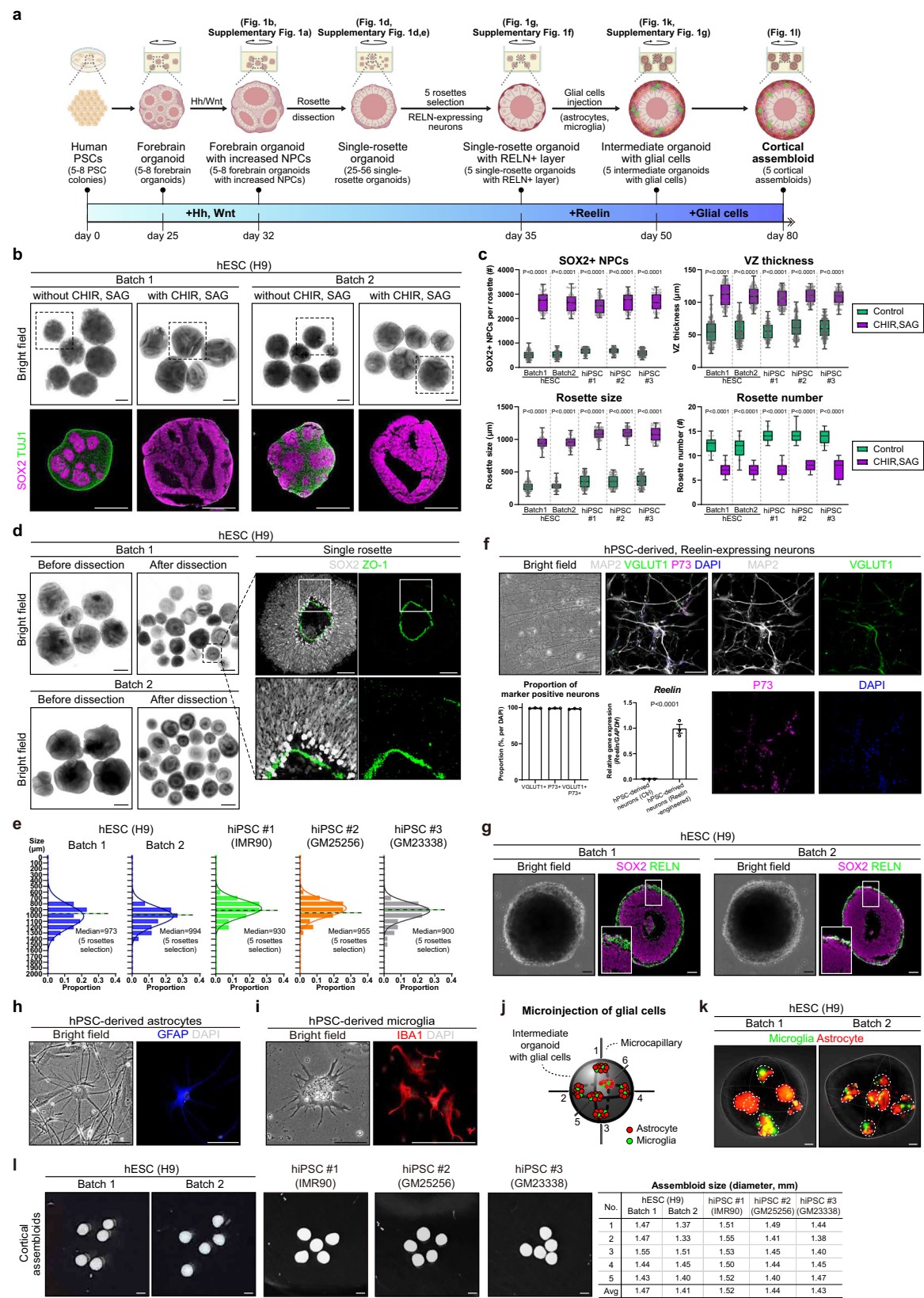

analysis of cortical assembloids using layer-specific cortical neuronal markers. We observed in the cortical assembloids a laminar distribution of layer-specific markers spanning from layer 6 to layer 1 in a proper sequential order (Fig. 2b and Supplementary Fig. 3a, b). Further quantitative analysis of the laminar expression of layer-specific neuronal markers revealed that the cortical assembloids are organized

into six stratified cortical layers (Fig. 2c and Supplementary Fig. 3a, b). These cortical layers consist of upper-layer neurons expressing SATB2 (layer 4), BRN2 (layer 3), CUX2 (layer 2), and RELN (layer 1), as well as deep-layer neurons expressing CTIP2 (layer 5) and TBR1 (layer 6). Interestingly, reconstituted neurons expressing RELN consistently remained on the outer surface of the cortical assembloids, serving as

**Fig. 1 | A stepwise, modular design of building uniform human cortical assembloids. a** Experimental scheme of creating cortical assembloids. Created in BioRender. Kim, Y. (2025) https://BioRender.com/jaxtaqw. **b** (top) Representative bright field images of early forebrain organoids (day 32), derived from hESCs, treated with CHIR99021 and SAG. (bottom) Magnified images of forebrain organoids, demarcated by dotted boxes on top panels, immunostained for NPCs (SOX2) and neurons (TUJ1). Scale bars, 1 mm. **c** Quantification of NPC populations (top left), VZ thickness (top right), rosette size (bottom left), and rosette number (bottom right) in early forebrain organoids. Center line, median; whiskers, min to max (show all points). **d** (left) Representative bright field images of manually-dissected, single rosettes. Scale bars, 1 mm. (right) Representative images of single rosettes, immediately after dissection at day 32, immunostained for SOX2 and ZO-1. Scale bars, 100 μm. **e** Quantification of the size of manually-dissected, single rosettes. Single rosettes were categorized by size based on the length of their diameters. The bar graphs show the relative proportion of the different-sized single rosettes. Curve, normal distribution; vertical line, median diameter. **f** Immunocytochemical analysis of hPSC-derived, reelin-expressing neurons. Scale bars, 100 μm. Quantification of the proportion of VGLUT1+, P73+, and VGLUT+/P73+ neurons are shown on the left. Relative gene expression of *RELN* is shown on the right. Data, mean values ± SEM. **g** Representative images of single rosettes encapsulated with the RELN+ layer (day 35) immunostained for SOX2 and RELN. Scale bars, 100 μm. Immunocytochemical analysis of astrocytes (**h**) and microglia (**i**) differentiated from hPSCs. Scale bars, 50 μm (**h**) and 25 μm (**i**). **j** Schematic illustration of the glial cell microinjection process. **k** Representative images of intermediate assembloids (day 50) immediately after being microinjected with glial cells. Astrocytes and microglia were labeled with RFP and GFP, respectively. Scale bars, 100 μm. **l** Representative images of cortical assembloids (day 80). Scale bars, 1 mm. Size measurements of individual cortical assembloids are shown on the right.

the first neuronal layer, which plays critical roles during the early developmental stages of the human brain (Fig. 2b, c and Supplementary Fig. 3a, b).

In addition to assessing neuronal maturity, we further evaluated the maturity of the cortical assembloids in terms of glial cell development. We found that glial cells, which were initially localized at the injection site on day 0 (Fig. 1k and Supplementary Fig. 1d), were observed to redistribute throughout the cortical assembloids during an extended culture period (Fig. 2d and Supplementary Fig. 4a). GFAP-expressing astrocytes exhibited high complexity and maturity, encompassing three morphologically distinct subtypes, including fibrous (8.5%), protoplasmic (44.0%), and interlaminar (47.5%) subtypes[31] (Supplementary Fig. 4b). In addition, these astrocytes formed tripartite synapses with neurons (Supplementary Fig. 4b). Moreover, IBA1-expressing microglia showed amoeboid to ramified morphologies, indicative of active and resting states, respectively (Supplementary Fig. 4c). These microglia were observed to be in close proximity to neurons and displayed active movement within cortical assembloids, indicative of functional activity such as the potential for engulfing and remodeling neuronal synapses (Supplementary Fig. 4c and Supplementary Movie 2).

Taken together, the structural characteristics of cortical assembloids suggest a comparable level of structural maturity to the developing human brain at later stages of brain development, wherein radially organized neurons establish the lamination of six cortical layers and glial cells, such as astrocytes and microglia, are integrated and organized with neurons to form functional networks within the cortex[18,32,33].

## Cortical assembloids represent spontaneous neural activity and functional connectivity

We next investigated the functional maturity of the cortical assembloids by examining the spontaneous neural activity and functional connectivity. We first assessed synapse formation in cortical assembloids and observed multiple puncta, marked by co-localization of the presynaptic marker VGLUT1 and the postsynaptic marker PSD95 in the cortical assembloids, indicating the presence of synapses (Fig. 3a, b and Supplementary Fig. 5a). To further evaluate the spontaneous neural activity of cortical assembloids, we performed calcium imaging analysis without external stimulation to monitor intracellular calcium dynamics in multiple neurons (Fig. 3c–e and Supplementary Fig. 5b). We observed spontaneous calcium surges in multiple individual neurons, occurring at an average rate of 3–4 surges per minute (Fig. 3c–e; Supplementary Fig. 5b and Supplementary Movies 3 and 4). In addition to calcium dynamics, we evaluated various electrophysiological properties of cortical assembloids by performing extracellular recordings using multi-electrode arrays (MEAs). Our analysis demonstrated robust electrical activities in cortical assembloids, characterized by spontaneous spikes with discrete action potentials and depolarization, as shown in the waveforms (Fig. 3f). Moreover, most of the firing events in cortical assembloids appeared as bursts, exhibiting repetitive and periodic patterns with an average burst frequency of 0.33 Hz and an interburst interval (IBI) of 3.04 s (Fig. 3g, h and Supplementary Fig. 5c). These data strongly suggest that cortical assembloids exhibit the robust and spontaneous neural activity.

To investigate whether these active neurons in cortical assembloids are functionally connected, we analyzed area-wide calcium spike patterns to evaluate the synchrony of calcium surges. We observed highly synchronized calcium activity across the entire area of the cortical assembloids at a bulk area level (Fig. 3c and Supplementary Movie 3). At the single-cell level, individual neurons displayed simultaneous calcium surges within short time windows (1–3 s) (Fig. 3c–e and Supplementary Movie 4), indicative of functionally connected neuronal networks. This synchronized pattern of neuronal activities was further confirmed through channel-wise MEA analysis. We observed highly synchronized neural activity across multiple channels, displaying network burst events in cortical assembloids (Fig. 3f–h and Supplementary Fig. 5c). Furthermore, these network bursts showed periodic and regular oscillatory patterns, with an average network burst frequency of 0.3 Hz (Fig. 3h and Supplementary Fig. 5c). These functional characteristics observed in cortical assembloids, such as synchronized and oscillatory network burst events, mirror the complex spatiotemporal dynamics seen in the developing human brain, wherein synchronous activity of oscillating networks is a prominent feature of functional cortical networks facilitating the coordinated propagation of synaptic signals between functionally related cortical areas[34–36].

Taken together, our data from structural and functional analysis demonstrate that the cortical assembloids represent a mature and interconnected six-layered cortical architecture with cellular diversity, along with the functional synapses and connected neuronal networks.

## Cortical assembloids recapitulate the cellular composition and transcriptome of the human developing brain at the single-cell level

To examine the maturity and cellular complexity of our cortical assembloids at the single-cell transcriptome level, we performed a comparative analysis of single-cell RNA sequencing (scRNA-seq) data between cortical assembloids and cortical organoids developed from widely utilized, four independent protocols[5,8,20,37] (Figs. 4 and 5; Supplementary Figs. 6 and 7 and Supplementary Data 1). We acquired the scRNA-seq datasets of 13 cortical assembloids derived from various experimental batches of human embryonic stem cell (hESC) and three different human induced pluripotent stem cell (hiPSC) lines. Additionally, we obtained publicly available scRNA-seq datasets of four independent, widely used cortical organoids developed using methods reported in previously published studies[5,8,20,37]. Single-cell datasets of cortical assembloids derived from different experimental batches and lines, as well as the datasets from four widely used cortical

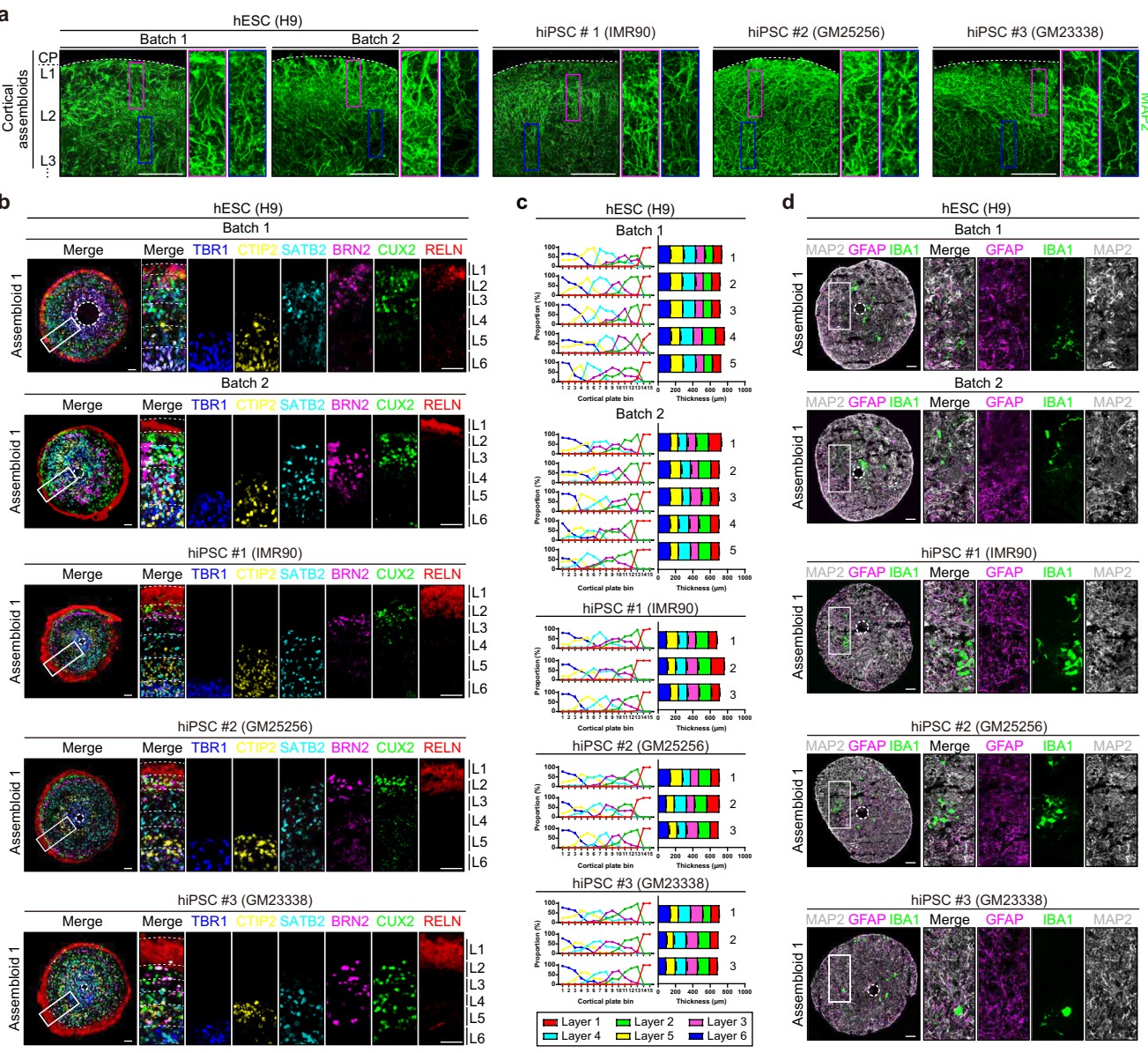

**Fig. 2 | Cortical assembloids represent the six-layered cortical structure and contain functional glial cells. a** Representative images of cortical assembloids (day 80) immunostained for MAP2 to visualize neuronal projections across cortical layers. CP cortical plate, L1 layer 1, L2 layer 2, L3 layer 3. Scale bars, 100 μm. **b** Representative images of 6-layered cortical structures of cortical assembloids (day 80). Left panels show merged images of three serial sections at 8-μm intervals in which each section was immunostained for TBR1/CTIP2, SATB2/RELN, and BRN2/CUX2, respectively. Dotted lines demarcate the border of each layer. L1 layer

1, L2 layer 2, L3 layer 3, L4 layer 4, L5 layer 5, L6 layer 6. Scale bars, 100 μm. **c** Quantification of the cortical thickness of each of the 6 layers. The six cortical layers were delineated based on the expression of specific markers associated with each layer of the cortex (TBR1 for layer 6, CTIP2 for layer 5, SATB2 for layer 4, BRN2 for layer 3, CUX2 for layer 2, and RELN for layer 1). The cortical thickness in individual assembloids is presented as independent bars, labeled with sequential numbers on the graph. **d** Representative images of cortical assembloids (day 80) for neurons (MAP2), astrocytes (GFAP), and microglia (IBA1). Scale bars, 100 μm.

organoids were integrated and processed for batch correction. Each dataset was then systematically clustered through principal component analysis (PCA) based on highly variable genes, which led to the identification of distinct clusters. The differential gene expression signatures of the identified clusters in each dataset were then analyzed to assign each cluster to pre-existing, endogenous cell types. This comprehensive approach enabled us to evaluate the maturity, including the cellular complexity, of cortical assembloids at the level of single-cell transcriptome, in comparison to previously reported organoids.

Our analysis showed that all cortical assembloids exhibited a high degree of consistency, containing seven transcriptionally distinct cell types within forebrain lineages, including radial glia (RG), dividing RG, intermediate progenitor cells (IPCs), excitatory neurons, a small

number of inhibitory neurons, astrocytes, and microglia (Fig. 4a–c; Supplementary Fig. 6 and Supplementary Data 1). Notably, analysis of cellular compositions and transcriptional profiles of cortical assembloids demonstrated a high correlation with the human fetal brain at late fetal periods (Fig. 4d). Further in-depth analysis revealed that cortical assembloids generated substantial populations of cells corresponding to each cortical layer, which align with the specific laminar organization of the human cortex[38,39] (Fig. 5a–c and Supplementary Fig. 7). Moreover, the trajectory analysis demonstrated that the excitatory neurons in each cortical layer of cortical assembloids followed a developmental sequence, generating layer-specific neurons in order from layer 6 to layer 2 (Fig. 5d), closely mirroring the temporal pattern observed in the developing human cortex[40]. This finding strongly suggests that cortical assembloids are comparable to the human

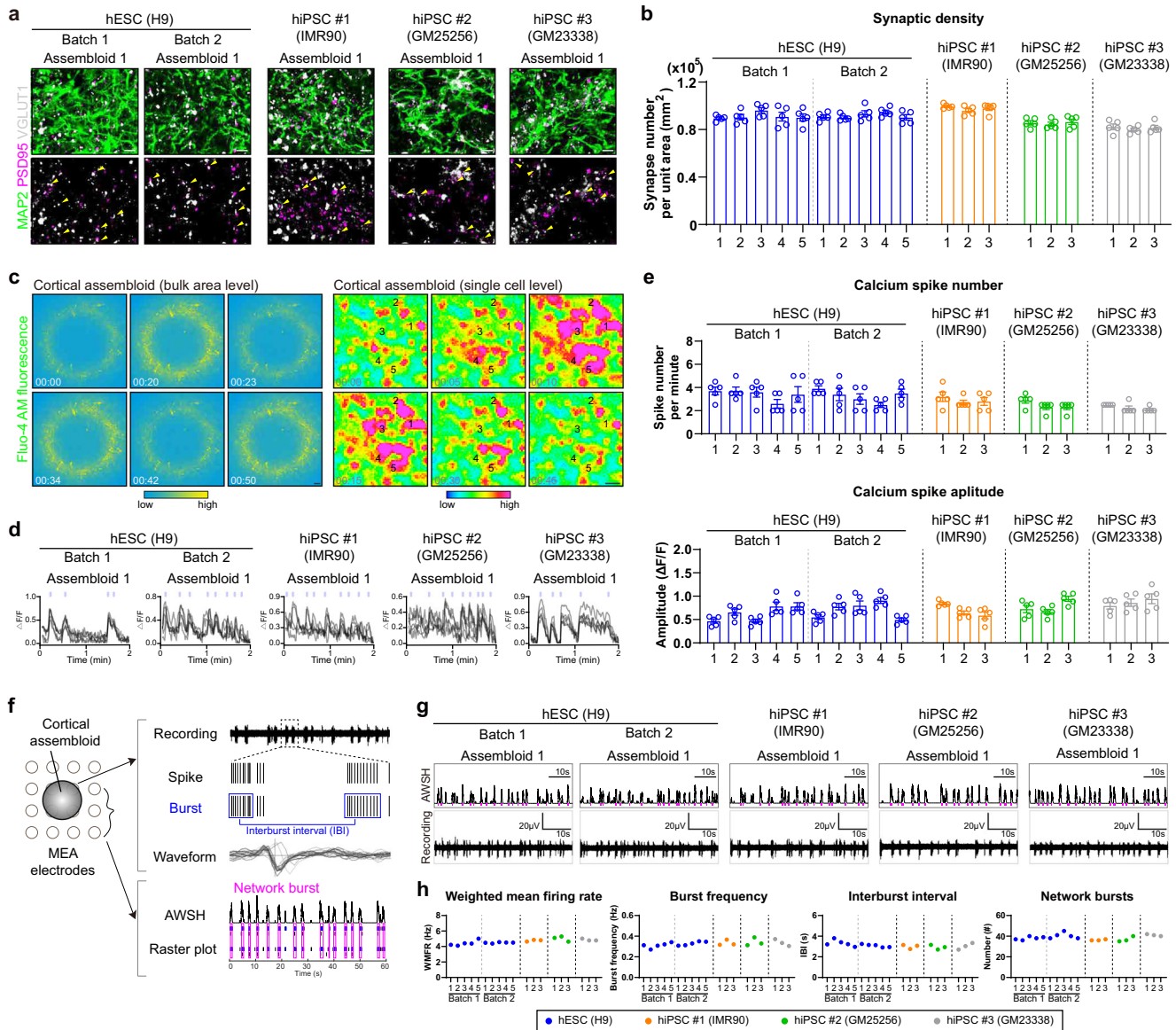

**Fig. 3 | Cortical assembloids represent the functional connectivity.**
**a** Representative images of cortical assembloids (day 80) immunostained for neurons (MAP2) and synapses (PSD95 and VGLUT1). The yellow arrowheads indicate synapses co-localized with PSD95 and VGLUT1. Scale bars, 50 μm.
**b** Quantification of synaptic density by calculating the number of synapses per unit area (mm²) in each section. The synaptic density in individual assembloids is presented as independent bars, labeled with sequential numbers on the X-axis. Data, mean values ± SEM. **c** (left) Representative images of calcium imaging analysis of cortical assembloids (day 80) at bulk area level. Scale bar, 100 μm. (right) Representative images of calcium imaging analysis of cortical assembloids (day 80) at single-cell level. Scale bar, 10 μm. **d** Representative images of calcium imaging analyses of five selected cells in cortical assembloids. **e** Quantification of spike number per minute (top) and amplitude (bottom) analyzed by calcium imaging of selected cells in cortical assembloids. The number of spikes and amplitude in

individual assembloids are presented as independent bars, labeled with sequential numbers on the X-axis. Data, mean values ± SEM. **f** Schematic representation illustrating the analysis of electrical activity from MEA recordings of cortical assembloids. Each bar represents a spike. A spike cluster (shown in blue) represents a burst. Burst occurring simultaneously across multiple channels is defined as a network burst (shown in magenta). AWSH, array-wide spike histogram.
**g** Representative images of the array-wide spike histogram (AWSH) and recording plot analyzed by MEA in cortical assembloids. Network bursts are highlighted in magenta. **h** Quantification of the weighted mean firing rate (WMFR), burst frequency, IBI, and the number of network bursts in cortical assembloids measured by MEA. Recordings were performed for 2 min. The WMFR, burst frequency, IBI, and the number of network bursts in individual assembloids are presented as independent dots, labeled with sequential numbers on the X-axis.

developing brain in terms of cortical layer specification and cellular compositions.

In comparison to cortical assembloids, analysis of scRNA-seq datasets from four independent, widely used cortical organoids reported in previous studies[5,8,20,37] revealed that, although the cell types and their proportions varied across protocols, these cortical organoids were primarily composed of RG, dividing RG, excitatory neurons and few inhibitory neurons, with a small proportion of IPCs as well as few other cell types (Fig. 4a–c; Supplementary Fig. 6 and

Supplementary Data 1). Specifically, cortical organoids developed using methods by Kadoshima, et al.[5] or Xiang, et al.[8] exhibited a high level of heterogeneity between batches, and predominantly consisted of excitatory neurons, RG, dividing RG, with few inhibitory neurons and astrocytes (Fig. 4a–c). The majority of neurons in these organoids showed characteristics typical of deep-layer neurons, comprising layer 5 and layer 6 neurons, with relatively fewer upper-layer neurons (layers 2, 3, and 4), and the absence of layer 1 neurons (Fig. 5a–c and Supplementary Fig. 7). Similar deep-layer properties were also

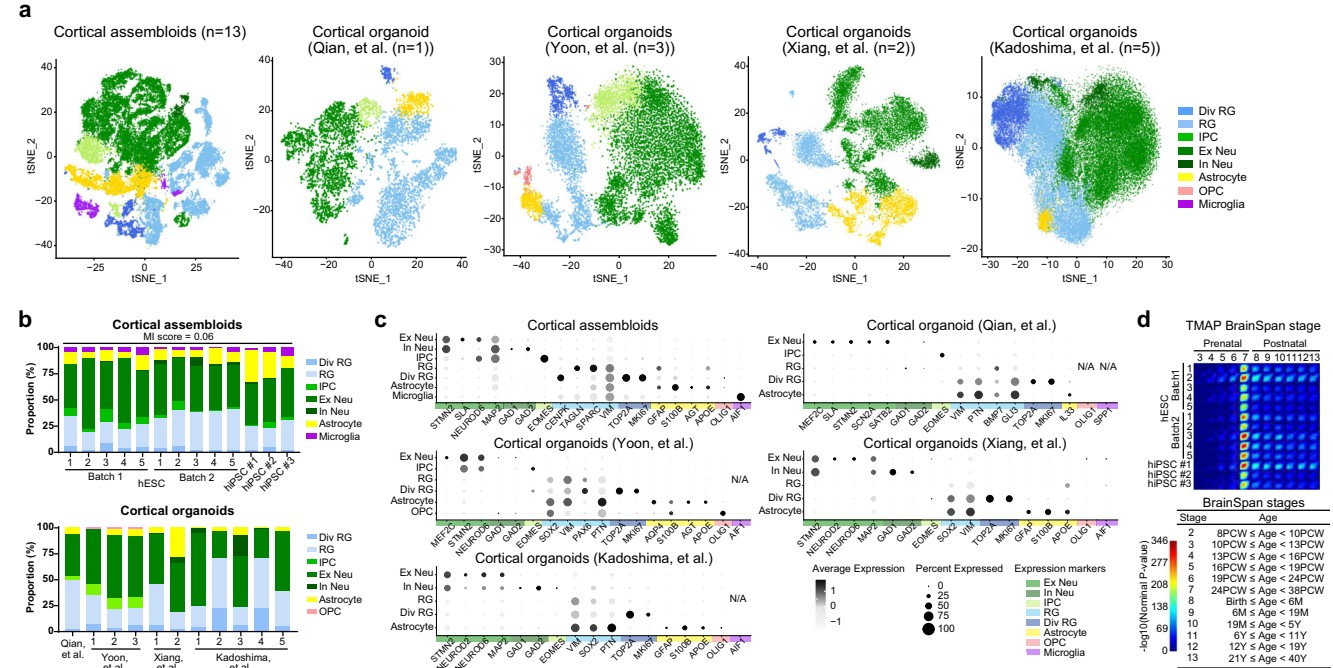

**Fig. 4 | Comparative scRNA sequencing analysis of cell composition and gene transcriptome of cortical assembloids. a** tSNE plots of scRNA-seq data from cortical assembloids and eleven independent, widely-utilized cortical organoids, developed using methods by Qian, et al.[20], Yoon, et al.[37], Xiang, et al.[8], and Kadoshima, et al.[5]. Cells are colored by cell type and labeled with cell type annotations. RG radial glia, Div RG dividing RG, IPC intermediate progenitor cell, Ex Neu excitatory neuron, In Neu inhibitory neuron, OPC oligodendrocyte progenitor cell. **b** Analysis of the proportion of individual cell types in cortical assembloids, as well as cortical organoids developed using four different protocols in previous studies. **c** Dot plots showing gene expression of the selected genes expressed in each cluster of current cortical organoids and cortical assembloids (circle size, proportion of cells; degree of shading, expression level). **d** Transition mapping (TMAP) of gene expression of cortical assembloids and human brain tissues from the BrainSpan dataset (compared to stage 2). BrainSpan stages and corresponding ages are shown below. PCW post conception weeks, M months, Y years.

observed in cortical organoids developed from Yoon, et al.[37], exhibiting a high proportion of early-born layer 6 neurons with a relatively lower proportions of upper-layer neurons (Fig. 5a–c and Supplementary Fig. 7). Excitatory neurons in cortical organoids developed from Qian, et al.[20] also exhibited deep-layer properties with 30 % of layer 6 neurons, alongside smaller proportions of upper-layer neurons, including 15 % of layer 1, 21 % of layer 3, and 16 % of layer 4 neurons, with the absence of layer 2 and 6 neurons (Fig. 5a–c and Supplementary Fig. 7). Furthermore, a minor proportion or absence of astrocytes was observed in all cortical organoids, except for one batch of the organoids developed from Xiang, et al.[8], which exhibited a high degree of heterogeneity between batches, with the complete absence of microglia. Notably, in contrast to cortical assembloids, the pseudotime trajectory analysis revealed that the excitatory neurons in each cortical layer of cortical organoids followed a developmental process that did not correspond to the sequential development of cortical layers observed during the human cortex development (Fig. 5d). Taken together, these data suggest that widely-used cortical organoids reported in previous studies consist of a high proportion of deep-layer neurons with relatively lower populations of late-born neurons, alongside a minor population or absence of glial cells such as astrocytes and microglia.

Comparative analysis was further performed to assess the level of maturity of cortical assembloids compared to previously reported cortical organoids. We found that our cortical assembloids contained a greater proportion of glial cells, including astrocytes and microglia. Specifically, astrocytes and microglia constituted approximately 13% and 4% of the total cell population, respectively, rendering cortical assembloids more comparable to the human developing brain than previously reported cortical organoids with lower proportions of astrocytes, as well as the complete absence of microglia (Fig. 5e). These

results strongly suggest that cortical assembloids exhibit enhanced maturity at the level of cellular compositions.

In addition, cortical assembloids exhibited all six distinct cortical layers with a reduced presence of deep layer neurons (layers 5 and 6) and a significantly higher proportion of upper layer neurons (layers 1, 2, 3, and 4) (Fig. 5f). Although their proportions varied across protocols, excitatory neurons in previously reported cortical organoids were predominantly composed of a deep-layer neurons (layers 5 and 6) with relatively lower levels of upper-layer neurons, alongside the absence of certain layer-specific upper-layer neurons, leading to an incomplete alignment with all six cortical layers (Fig. 5b, f).

Lastly, mutual information (MI) score[24] was calculated across samples to evaluate the variability of our cortical assembloids derived from multiple batches and different lines, revealing that all cortical assembloids demonstrated a high level of consistency (Fig. 4b). In contrast, currently available cortical organoids developed using the same protocols exhibited a comparatively higher degree of variability between batches and individual organoids within the same methodology, as demonstrated by the variance test (Fig. 5g). These data suggest high reproducibility and low variation of cortical assembloids across batches and lines, further implying a high level of consistency and uniformity without significant batch effects and heterogeneity.

To further assess the maturity and consistency of our cortical assembloids in comparison with existing brain organoids—extending beyond the four most commonly used protocols we evaluated above—we expanded our comparative scRNA-seq analysis by incorporating datasets from the recently published human neural organoid cell atlas (HNOCA)[25]. This study includes 36 brain organoid scRNA-seq datasets, 16 of which specifically target the cerebral cortex region. Among these, we excluded nine datasets for organoids generated from four widely utilized protocols already analyzed, leaving seven

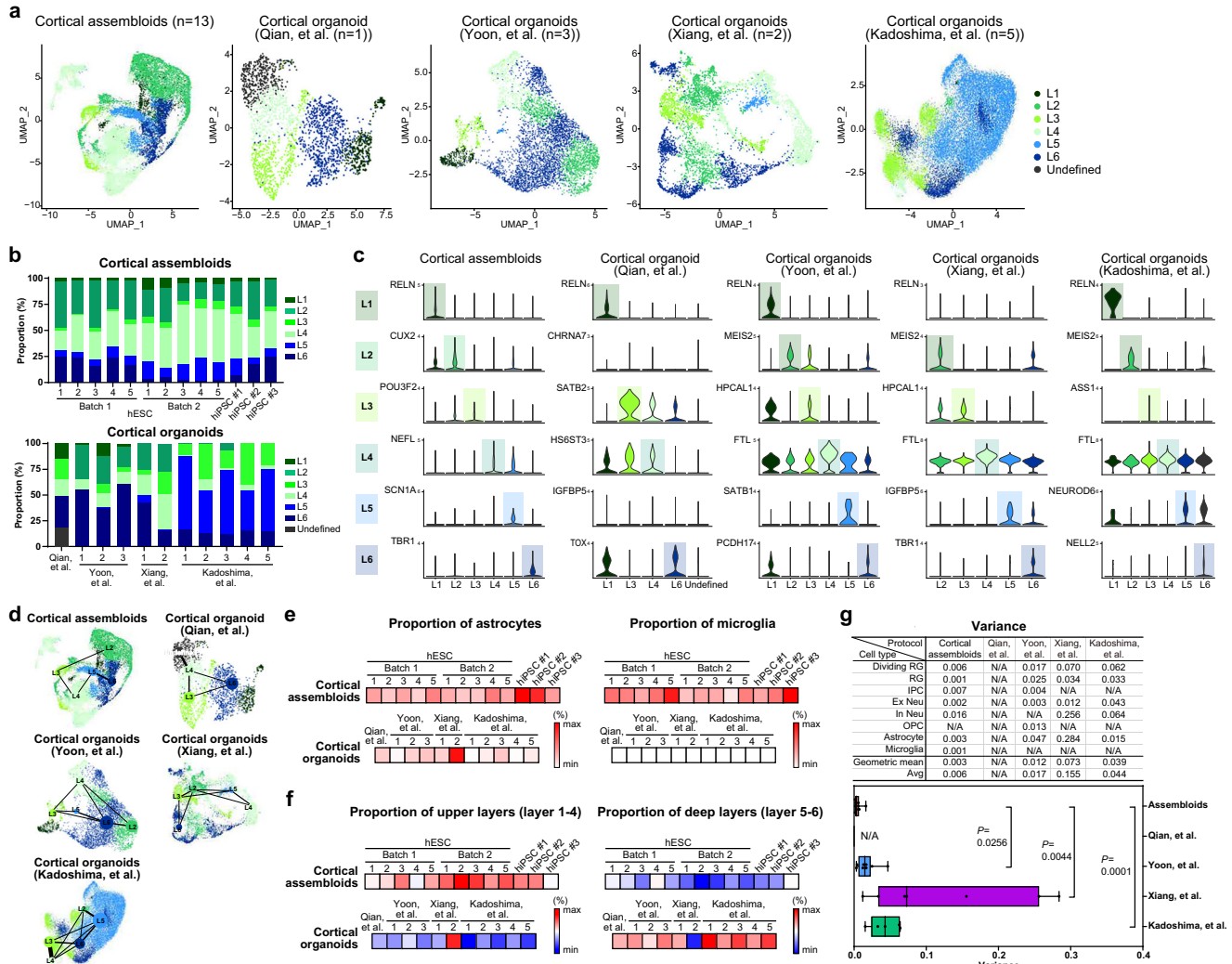

**Fig. 5 | Cortical assembloids represent the distinct neuronal cell populations of the six cortical layers of the human brain.** **a** UMAP plots of scRNA-seq data for clusters marked as excitatory neurons in Fig. 4a, derived from publicly available current cortical organoids and cortical assembloids. L1, layer 1; L2, layer 2; L3, layer 3; L4, layer 4; L5, layer 5; L6, layer 6. **b** Analysis of the proportion of individual cells according to the six cortical layers in cortical assembloids as well as cortical organoids developed using four different protocols in previous studies. **c** Violin plots showing the selected marker genes expressed in each of the six cortical layers of current cortical organoids and cortical assembloids shown in (**a**). **d** Trajectory analysis for excitatory neurons within current cortical organoids and cortical assembloids. Cells are colored by six cortical layers. **e** Comparative analysis of the

proportion of astrocytes and microglia in cortical assembloids. The proportions of astrocytes and microglia in cortical assembloids, in comparison to those in cortical organoids, are visualized through heatmaps. **f** Comparative analysis of the proportion of upper-layers and deep-layers in cortical assembloids. The proportions of upper-layers (layer 1–4) and deep-layers (layer 5–6) in cortical assembloids, in comparison to those of cortical organoids, are visualized through heatmaps. **g** The variance of each cell type between individual cortical assembloids, as well as among individual cortical organoids developed using four different protocols. Each data point represents the variance of a given cell type within the indicated dataset. Center line, median; whiskers, min to max (show all points).

new datasets[4,22,41–45] for comparison. We then performed additional comparative scRNA-seq using these seven new datasets and compared the results with our assembloid datasets (Supplementary Figs. 8–10 and Supplementary Data 1).

In line with our findings, these additional analyses confirmed that our cortical assembloids exhibited a high degree of maturity, including increased numbers of upper-layer neurons and glial cells, as well as the formation of six distinct cortical layers (Supplementary Figs. 8–10). Furthermore, variance analysis indicated that cortical assembloids exhibited a higher level of consistency compared to existing brain organoids, which demonstrated greater variability among individual organoids generated using the same methodology (Supplementary Fig. 8g). Additionally, we performed a comparative scRNA-seq analysis by integrating our cortical assembloid dataset with datasets from

eleven recently published cortical organoid models as well as the developing human brain. These integrated analyses further confirm that our assembloids exhibit a high degree of consistency and maturation (Supplementary Fig. 11a–h and Supplementary Data 1), aligning with findings from the analyses of individual datasets (Figs. 4 and 5 and Supplementary Figs. 6 and 7), and closely resemble the second to third trimester human fetal cortex in cell-type composition and transcriptomic profiles (Supplementary Fig. 11i–m and Supplementary Data 2).

Taken together, these comprehensive scRNA-seq analyses demonstrate that cortical assembloids comprise the six distinct cortical layers of neurons and glial cells with a high degree of consistency, suggesting a level of maturity and cellular complexity highly comparable to that of the cortex in the developing human brain (Figs. 4 and 5 and Supplementary Figs. 6–11).

## Discussion

A major conceptual advance presented in our current work is the development of effective strategies to generate reconstituted cortical organoids called cortical assembloids, which are defined as "organoids created by reconstituting multiple cell types of human tissues"[23,46], as opposed to the alternative definition "fusion/hybrid organoids"[4,47]. Our reconstitution technology for creating cortical assembloids differs conceptually from fusion-based approaches that combine region-specific organoids to promote inter-tissue connectivity. It also diverges from "chimeroids," which are generated by mixing human PSCs or neural stem cells (NSCs) from different donors to create chimeric cell populations that are then co-cultured and differentiated into 3D structures, resulting in chimeric organoids[48]. In contrast, our method integrates multiple cell types and tissue layers within a single organoid structure, allowing us to prioritize "intra-tissue maturity and cellular diversity" rather than inter-tissue connectivity. As a result, our cortical assembloids display enhanced structural integrity and cortical specificity.

Our methodology offers three major technical innovations. First, we developed the single-rosette technique to generate consistent cortical assembloids with single VZs. In current brain organoid technologies, major heterogeneity issues mainly arise from the difficulty in controlling the number and the size of each rosette in conventional multi-rosette organoids. We addressed this by manually dissecting early-stage brain organoids to generate single rosette organoids, thereby achieving uniformity and minimizing variation between organoids and different batches. Two critical signaling molecules involved in embryonic patterning, Hedgehog and Wnt, were employed to increase neural progenitor populations at an early stage of organoid formation, based on previous findings that these two patterning cues induce cellular proliferation of NPCs, leading to the formation of embryonal tumors with multilayered rosettes[27,28]. This increased the NPC population to create larger rosettes and cortical layers via activation of the Hh and Wnt pathways. Although several previous papers have described single-rosette-based protocols for developing brain organoids[49-53], these approaches mostly relied on two-dimensional (2D), premature structures developed through 2D neuroepithelial culture. These 2D-derived single neuroepithelial structures either fail to form 3D rosettes with proper structural integrity or develop into multi-rosette organoids during subsequent 3D culture, with only a small proportion of organoids forming single-rosette structures—resulting in high batch-to-batch variability across individual organoids. These challenges in establishing uniform, properly organized, and mature single neuroepithelial structures likely stem from the inherent difficulty in controlling the size of vesicle-like neuroepithelial structures during early development, which substantially contributes to issues of reproducibility and heterogeneity. A technical advance of our study over these earlier works lies in utilizing 3D, uniform rosettes dissected from 3D brain organoid culture at the early stage, which yields not only uniform but also highly reproducible assembloids, eliminating batch effects and heterogeneity.

Second, we developed the encapsulation method to achieve the maturity of six cortical layers in cortical organoids. By encapsulating uniform single-rosette organoids—each containing a single lumen that later becomes a VZ—with the first layer of RELN-expressing neurons, we achieved a high level of experimental control and the formation of mature cortical layers in cortical organoids. This approach established a uniform interaction between the entire NPC layer and the first layer of neurons expressing RELN, thereby initiating the consistent patterning of cortical layers in the correct order.

Lastly, our reconstitution methodology incorporates neurons, astrocytes, and microglia, creating assembloids with greater cellular diversity. These assembloids demonstrate functional connectivity and dynamic cellular interactions between neurons and glial cells, which

are critical for studying the complex cellular interplay seen in human neurodevelopmental disorders.

One limitation of our cortical assembloids is the production of a relatively small number of inhibitory neurons (Fig. 4). Our method employs dorsally patterned cortical organoids, which predominantly generate excitatory neurons while producing a small population of interneurons. This characteristic is consistent with many existing cortical organoids that recapitulate the intrinsic developmental programs of the dorsal forebrain, where inhibitory interneurons are not readily produced[20,37,54]. Therefore, the heightened neuronal activity observed in our cortical assembloids (Fig. 3 and Supplementary Fig. 5) is likely driven by increased excitatory neuron maturity, promoted by our module-based cellular reconstitution approach. Although several studies have reported the presence of inhibitory neurons in cortical organoids, these typically result from inconsistent and uncontrolled ventral patterning, at the expense of proper dorsal specification and subsequent cortical maturation, occasionally leading to the sporadic formation of inhibitory neurons[24,50,55]. In a few other cases, inhibitory neurons are incorporated into cortical organoids by fusing dorsally patterned cortical organoids with ventral forebrain organoids. However, due to the inherent limitations of the fusion-based approach, these organoids exhibit substantial variability across experiments, caused by uncontrolled and unpredictable migration of interneurons[4,8,9]. Further studies will be necessary to explore experimental strategies for incorporating an increased number of inhibitory neurons into cortical assembloids, such as refining differentiation protocols to promote the development of inhibitory neurons within cortical organoids or developing more controlled and reproducible fusion approaches.

Another limitation of our technology is its limited scalability. The inherent complexity of our modular, stepwise method may not be optimal for large-scale applications or high-throughput screening. Nevertheless, our approach offers a reliable strategy for generating high-quality, reproducible cortical organoids with enhanced uniformity, making it a valuable platform for disease modeling and mechanistic studies that require consistent and mature organoids. Future studies will be necessary to develop additional approaches that can be reliably scaled to meet the demands of high-volume production.

In summary, we developed stepwise reconstitution techniques, including the early-stage generation of single-rosette organoids and the subsequent spatial reconstitution of RELN-expressing neuronal layers and non-neuronal glial cells—an approach not explored in fusion-based methods. Our study produces uniform, mature cortical assembloids with enhanced cellular diversity, advancing the development of innovative organoid models that capture the mature, functional characteristics of the human brain, including the dynamic interplay between neurons and glial cells. We believe our technology offers a powerful platform for investigating complex cellular interactions involved in brain disorders and for developing new, patient-specific therapeutic options.

## Methods

### hPSC culture

hESCs (H9) and hiPSCs (IMR90, female) were obtained from WiCell. hiPSCs (GM25256, male; GM23338, male) were obtained from the National Institute of General Medical Sciences Cell Repository through the Coriell Institute for Medical Research. All hPSCs were maintained on mitomycin C-treated mouse embryonic fibroblasts (MEFs) in hPSC medium, containing DMEM/F12 (Gibco) supplemented with 20 % KnockOut Serum Replacement (Gibco), 1× Glutamax (Gibco), 1× Non-essential amino acids (Gibco), 1 % penicillin–streptomycin, 100 µM 2-Mercaptoethanol (Sigma), and 10 ng/mL human bFGF (Peprotech). All hPSC lines were cultured on each well of a 24-well plate with 0.5 mL

of culture media (with $5 \times 10^4$ feeder cells per well). The cells were fed daily and passaged by manual dissection at 70 % confluence (10 manually-dissected, small colonies of hPSCs were plated on each well of a 24-well plate). Cells used in this study were negative for mycoplasma contamination (e-Myco Mycoplasma PCR detection kit). All work using hPSC lines was performed in accordance with the ISSCR Guidelines for Stem Cell Research and Clinical Translation (2021).

### In vitro differentiation of neurons, astrocytes, and microglia

Glutamatergic neurons were differentiated from hPSCs using the protocol modified from the previous work[56]. In brief, hPSC colonies were detached from the feeder layer using collagenase IV (Thermo) at 37 °C for 1 h. 10-16 PSC colonies (5–8 hPSC colonies from each well of 24-well plate, 2 wells) were collected in 0.5 mL of N2/B27 media, consisting of DMEM/F12, 1× N2 supplement (Gibco), 1× B27-RA supplement (Gibco), and 1× penicillin–streptomycin, and plated on 35 mm petri dish. On day 1, the medium was changed with N2/B27 medium supplemented with 10 μM SB431542 (Sigma) and 0.1 μM LDN193189 (Stemgent). On day 7, EBs were transferred to Matrigel (Growth Factor Reduced, Corning)-coated (10 % Matrigel in DMEM/F12) plates and cultured in N2/B27 medium with 10 nM SB431542. On day 16, cells with rosette structures were collected and dissociated mechanically by pipetting. Dissociated cells were plated on Matrigel-coated plates in NPC medium, consisting of DMEM/F12, 1× N2 supplement, 1× B27-RA supplement, 20 ng/ml bFGF, and 1 μg/mL laminin (Thermo). At 80 % confluency, NPC medium was changed into neuronal medium, consisting of DMEM/F12, 1× N2 supplement, 1× B27-RA supplement, 1× penicillin–streptomycin, 20 ng/mL BDNF (Peprotech), 10 ng/mL GDNF (Peprotech), 250 μg/mL dibutyryl cyclic-AMP (Biogems), and 200 nM L-ascorbic acid (Sigma). The cultures were maintained in neuronal medium for 2 weeks on 10 μg/mL poly-L-ornithine (Sigma) and 5 μg/mL laminin-coated plates. Neuronal identity was validated by RT-qPCR and immunocytochemistry.

hPSC-derived astrocytes were generated as previously described[30]. Briefly, hPSC-derived NPCs were differentiated into astrocytes by seeding dissociated single cells at 15,000 cells/cm² on Matrigel-coated plates in Astrocyte medium (ScienCell: 2 % FBS, Astrocyte Growth Supplement, 1 % penicillin–streptomycin in astrocyte basal medium) and maintained. After 30–40 days of differentiation, hPSC-derived astrocytes were characterized using RT-qPCR and immunocytochemistry.

hPSC-derived microglia were generated as previously described[29]. Briefly, hPSC colonies were detached from the feeder layer with collagenase IV for 1 h. hPSC colonies were collected in microglia medium consisting of 10 ng/mL IL-34 (Peprotech) and 10 ng/mL GM-CSF (Peprotech) in 100-mm sterile petri dishes. After 7–14 days of differentiation, EBs with a cystic morphology were selected and transferred to 10 μg/mL poly-L-ornithine and 5 μg/mL laminin-coated plates. Six additional titrations were performed every 5 days. Further maintenance was performed in microglia medium with 100 ng/mL IL-34 and 5 ng/mL GM-CSF. Microglial identity was validated by RT-qPCR and immunocytochemistry.

### Generation of RELN-expressing neurons

The lentiviral construct for RELN expression was generated by subcloning domains 3–6 (central domains) of *RELN* from the pCrl construct (Addgene #122443, kindly gifted by Mi-Ryoung Song, Gwangju Institute of Science and Technology) into the pLenti 6.3-DEST (Thermo) lentiviral expression vector. NPCs were transduced with the *RELN*-containing lentivirus using protamine sulfate (5 μg/mL). Three days after transduction, NPCs were expanded in NPC medium, containing blasticidin (6 μg/mL), for antibiotic selection. NPCs were then differentiated into glutamatergic neurons. RT-qPCR was performed to confirm RELN expression.

### A stepwise development of cortical assembloids

hPSCs were cultured on a feeder layer in a 24-well plate (5–8 hPSC colonies in each well of the 24-well plate). When the diameter of hPSC colonies reached 1.0–1.5 mm, the hPSC medium was replaced with 0.25 mL of 1 mg/mL collagenase IV and incubated at 37 °C for 1–2 h to detach the colonies from the feeder layer without disrupting their colony structure. Detached colonies were transferred to a 15-mL tube and washed with 1 mL of hPSC medium. 2 mL of EB medium, comprising hPSC medium (without bFGF) supplemented with 2 μM Dorsomorphin (Sigma) and 2 μM A83-01 (Tocris), was added, and the 5–8 hPSC colonies (from each well of the 24-well plate) in EB medium were transferred and cultured in a 35-mm petri dish at 37 °C to form EBs. On day 5, half of the medium was replaced with neural induction medium consisting of DMEM/F12, supplemented with 1× N2 supplement, 10 μg/mL Heparin (Sigma), 1× penicillin–streptomycin, 1× Non-essential amino acids, 1× Glutamax, 1 μM CHIR99021 (Tocris), and 1 μM SB431542. On day 7, 5–8 EBs were transferred to a 1.5 mL microcentrifuge tube and mixed with 50 μL of Matrigel and 33 μL of neural induction medium. A total of 88 μL of mixture containing EBs was plated onto the center of a 35-mm petri dish and incubated at 37 °C for 30 min. Then, 2 mL of neural induction medium was added, and EBs were cultured until day 14 to induce neuroepithelium-like structures. On day 14, the Matrigel-embedded neuroepithelium structures were mechanically dissociated by gentle pipetting, and the resulting structures were then transferred to a new 35-mm petri dish containing 2 mL of differentiation medium consisting of DMEM/F12 supplemented with 1× N2 supplement, 1× B27 supplement, 1× penicillin–streptomycin, 1× 2-Mercaptoethanol, 1× Non-essential amino acids, 1× Glutamax, and 2.5 μg/mL Insulin (Sigma). These neuroepithelium structures were cultured in differentiation medium until day 25 in a shaking incubator (Eppendorf, New Brunswick S41i, 90 rpm) to form forebrain organoids with multiple rosettes.

From days 25 to 32, the organoids were cultured in differentiation medium supplemented with 1 μM CHIR99021 and 400 nM SAG (Millipore). Rosette structures in 32-day-old forebrain organoids were observed under an inverted microscope at 10X magnification (Evos XL Core, Invitrogen), and manually dissected using fine forceps (Fine Science Tools). To isolate single rosettes, the organoids were initially cut along the rosette boundaries into roughly 2–3 pieces, and each piece was then further dissected following the rosette structure to obtain individual rosettes. The edges of the dissected rosette recovered within a day of shaking culture, and organoids exhibiting a single, clear, round rosette were classified as single-rosette organoids suitable for further processing. About 5–7 single-rosette organoids were generated from one forebrain organoid. Since each batch of hPSCs typically yields approximately 5–8 forebrain organoids, this results in the generation of approximately 25–56 single-rosette organoids per batch. After dissection, the single-rosette organoids were cultured for an additional 3 days to stabilize the dissected structures.

On day 35, five single-rosette organoids, whose diameters were closest to the median values within the range of 25–56 single-rosette organoids, were selected for further procedures. For hESC-derived assembloids, two batches were used in each set of experiments (a total of 10 single-rosette organoids were generated, and 10 final assembloids were analyzed in each experiment). For hiPSC-derived assembloids, one batch was used (a total of 5 single-rosette organoids were generated, and 3 final assembloids were analyzed in each experiment). The resulting single-rosette organoids were then encapsulated with 1–2 μL of Matrigel containing RELN-expressing neurons at a concentration of $1 \times 10^4$ cells/μL and cultured in 2 mL of differentiation medium until day 50.

On day 50, hPSC-derived astrocytes ($1 \times 10^4$ cells) and microglia ($2 \times 10^3$ cells)[38] were then microinjected into the outer cortical layer of 50-day-old cortical assembloids using a microinjector (FemtoJet 4i, Eppendorf). Alternatively, aspirator tube assemblies for calibrated

microcapillary pipettes (Merck) can be used for microinjection, with one microcapillary employed to stabilize the organoids while another, loaded with glial cells, administers the injection. Cells were microinjected at six different locations, which are evenly distributed through the outer cortical layer, with 50 μm inside the surface of the cortical layer. The resulting assembloids were cultured for an additional 30 days in 2 mL of maturation medium consisting of Neurobasal medium (Gibco) supplemented with 1× B27 supplement, 1× penicillin–streptomycin, 1× 2-Mercaptoethanol, 0.2 mM Ascorbic Acid, 20 ng/mL BDNF, 20 ng/mL GDNF, and 0.5 mM cAMP (Sigma) to generate the final form of cortical assembloids. From days 14 to 80, all cultures were maintained in a shaking incubator with medium changes every other day.

## Lentivirus production
Lentivirus production was performed as previously described[23]. In brief, transfection mixtures were prepared by mixing 9 μg of packaging vector (gag, pol; pCMV.dR 8.74), 3 μg of envelope vector (VSV-G; pMD2.G), 10 μg of the transfer vector of interest, and three volumes of TransIT-LT1 Transfection Reagent (Mirus). 48 h after transfection, lentivirus-containing supernatants were collected and filtered through a 0.45-μm filter. The virus-containing supernatants were further concentrated by centrifuging at $7.1 \times 10^4 \times g$ (24,000 rpm) for 2 h at 4 °C.

## Lentiviral infection in cortical assembloids
Cortical assembloids were incubated in assembloid medium (2 mL) containing EGFP lentivirus ($1.0 \times 10^6$ TU/mL, 200 μL) with polybrene (10 μg/mL) for 3 h at 37 °C. The virus-containing medium was removed, and assembloids were washed with warm DPBS twice. Assembloids were cultured for 3 days more in assembloid medium and analyzed by fluorescence imaging.

## RT-qPCR
Total RNA was extracted from multiple cells and organoids as previously described[23]. In brief, cells and organoids were homogenized by trituration and trypsinization. RNA was extracted using the RNeasy Plus Mini Kit (QIAGEN) and first-strand cDNA was synthesised using a High-Capacity cDNA Reverse Transcriptase Kit (Applied Biosystems) with oligo dT. RT-qPCR was performed using SYBR Green Supermix (Applied Biosystems) and a One-step Cycler (Applied Biosystems). Gene expression was normalized to the housekeeping gene *GAPDH*. Primer sequences used in this study are as follows:

- *PAX6*: (F) GAATCAGAGAAGACAGGCCA / (R) GTGTAGGTATCATAACTCCG
- *TBR1*: (F) CTCAGTTCATCGCCGTCACC / (R) AGCCGGTGTAGATCGTGTCATA
- *DLX1*: (F) TGGAGGACCCAGGTCAAGAT / (R) TAGCTTCTTGGTGCGCTGAA
- *DLX2*: (F) ACGTCCCTTACTCCGCCAAG / (R) AGTAGATGGTGCGGGGTTTCC
- *LHX6*: (F) CTGGAAGCATGAGAACGCCG / (R) CGCGCAACAGCAGGGAAATA
- *NR2F2:* (F) AATGCTACATCCCGCCCACAG / (R) CGCGCAACAGCAGGGAAATA
- *GAPDH:* (F) GTCTCCTCTGACTTCAACAGCG / (R) ACCACCCTGTTGCTGTAGCCAA

## Immunohistochemistry
Immunohistochemistry was performed as previously described[23]. In brief, samples were fixed in 4 % paraformaldehyde (PFA) for 15 min and cryopreserved in 30 % sucrose overnight. Samples were embedded in an OCT compound (Sakura) and frozen at −20 °C. Then, 8–20-μm-thick sections were generated using a cryostat (Leica). Frozen sections were fixed in 4 % PFA for 20 min at 4 °C, washed with PBS three times, and blocked in 2 % goat serum and PBS containing 0.25% Triton X-100

(PBS-T) for 1 h at RT. The sections were then incubated with primary antibodies diluted in blocking buffer overnight at 4 °C. The following primary antibodies were used: TUJ1 (1:300, BioLegend, #801201), SOX2 (1:300, Millipore, #ab5603), CTIP2 (1:300, Abcam, #ab18465), CUX2 (1:300, Abcam, #ab216588), SATB2 (1:300, Abcam, #ab34735), TBR1 (1:300, Abcam, #ab31940), MAP2 (1:300, Abcam, #ab5392), GFAP (1:300, Dako, #Z0334), IBAI (1:60, Santacruz, #sc-32725), RELN (1:200, MBL, #D223-3), BRN2 (1:300, Santacruz, #sc-393324), PSD95 (1:300, Invitrogen, #51-6900), VGLUT1 (1:100, Santacruz, #sc-377425), ZO-1 (1:200, Santacruz, #sc-33725), P73 (1:200, Thermo, #PA5-28931), and PAX6 (1:30, DSHB, #AB_528427). Sections were washed three times with 0.25% PBS-T and incubated with secondary antibodies (1:1,000, Invitrogen) diluted in blocking buffer for 1 h at RT. The sections were washed with 0.25% PBS-T and mounted with Prolong Gold mounting reagent (Invitrogen).

For immunocytochemistry, cells were plated on 10 μg/mL poly-L-ornithine and 5 μg/mL laminin-coated coverslips on a 12-well plate. When reached 80 % confluence, cells were washed with PBS and fixed in 4% PFA for 5 min at RT. Cells were washed with PBS three times and blocked for 40 min at RT. Cells were then incubated with diluted primary antibodies for 1 h at RT and washed three times with PBS-T. Cells were then incubated with secondary antibodies diluted in blocking buffer for 40 min at RT. The cells were washed two times with PBS-T and mounted on glass slides.

## MEA recording
24-well MEA plates (Axion Biosystems, Atlanta, GA, USA) were coated with 10 μg/mL poly-L-ornithine and 5 μg/mL laminin solution prior to assembloid culture. Assembloids were placed on MEA plates and cultured for 2 weeks to be attached to the electrodes with media changes in every 3 days. After two weeks of attachment and stabilization, recordings were captured to measure basic parameters such as weighted mean firing rate, burst frequency, mean IBI, and the number of network bursts using a Maestro MEA system and AxIS Software Spontaneous Neural Configuration (Axion Biosystems). Spike detection was carried out using the AxIS software by setting an adaptive threshold at 5.5 times the noise's standard deviation estimated for each channel (electrode). Before recording, the plate was left to stabilize for 10 min inside the Maestro device.

The electrodes with at least 5 spikes/min were defined as active electrodes. Bursts within the data from each electrode were recognized based on an inter-spike interval (ISI) threshold requiring a minimum number of 5 spikes with a maximum ISI of 100 ms. A minimum of 12 spikes under the same ISI criterion, with a minimum of 50% active electrodes, was required for network bursts in the well. Raster plot and array-wide spike histogram were obtained using Axion Biosystems' Neural Metrics Tool.

## Whole-cell patch-clamp recording
Whole-cell patch-clamp recording was performed from sections sectioned from cortical assembloids. AAV1-hSyn1-GFP (Addgene #50465) was diluted in DPBS on ice and microinjected five times into the cortical assembloids. Assembloids were embedded in 4 % low-melting-point agarose and sliced into 200 μm-thick sections on a vibratome. Slices were recovered in oxygenated artificial cerebrospinal fluid (aCSF) containing 119 mM NaCl, 2.5 mM KCl, 2.5 mM CaCl₂, 2 mM MgSO₄, 1.25 mM NaH₂PO₄, 26 mM NaHCO₃ and 10 mM D-glucose for 1 h, equilibrated with 95% O₂ and 5% CO₂ (pH 7.3–7.4) at RT. After recovery, slices were transferred into a recording chamber perfused with aCSF solution. Whole-cell post-synaptic patch-clamp recordings were performed using a MultiClamp 700B amplifier (Molecular Devices). Recording glass pipettes (4–9 MΩ) were filled with an internal solution containing 120 mM K-Gluconate, 5 mM NaCl, 0.2 mM EGTA, 1 mM MgCl₂, 10 mM HEPES, 2 mM MgATP, and 0.2 mM NaGTP (adjusted to pH 7.2 with KOH). All the recorded cells showed GFP

expression. Membrane properties and excitability were measured and analyzed with Clampfit 10.1 software (Molecular Devices).

## Calcium imaging
Cortical assembloids were incubated with organoid medium containing 1 µM Fluo4-AM (Invitrogen) for 3 h at 37 °C. Assembloids were washed once with DPBS and incubated in brain organoid medium. Time-lapse image sequences were acquired for 2 min with 1 s intervals on a Nikon confocal microscope. $\Delta F/F$ traces in the selected cells were calculated and shown in the graph ($\Delta F/F = (F - F_0)/F$, where $F$ is the fluorescence at given time point and $F_0$ is the minimum fluorescence of each cell). The average amplitudes ($\Delta F/F$) and frequencies of spikes detected in the cortical assembloids are 0.6–1.4 and 3–4 spikes/min. Spontaneous calcium activities were analyzed with ImageJ software.

## scRNA sequencing
**Cell harvest.** Cortical assembloids were collected and washed twice with DPBS. The cells were treated with Accutase for 2–3 min at RT before being dissociated into single cells by pipetting and centrifugation at $300 \times g$ for 5 min. The cells were resuspended in the medium (DMEM with 10 % FBS) and strained through a 40-µm cell strainer. Each sample was run on 10× Chromium Single-Cell Chips (10x Genomics) following the manufacturer's instructions.

**scRNA library preparation and sequencing.** The scRNA-seq library was prepared using the Chromium Single Cell 3 Prime platform (v3.1 Chemistry, 10x Genomics) according to the manufacturer's instructions. Briefly, 10,000 cells per sample were loaded into the Chromium Controller in order to generate single Gel Bead-in-Emulsions (GEMs) with the Chromium Next GEM Single Cell 3 Prime Reagent Kit v3.1 (PN-1000268, 10x Genomics). The cells were lysed, and released RNAs were synthesized into cDNA through reverse transcription in individual GEMs. Full-length cDNA was synthesized by capturing polyadenylated mRNA with poly(dT) primers and barcoding it. cDNA was synthesized by incubating at 53 °C for 45 min and 85 °C for 5 min. cDNA amplification was performed for 12 PCR cycles following GEM cleanup to generate sufficient amounts of DNA for library construction. Single Cell 3' GEX and feature barcode libraries were sequenced on the Illumina NovaSeq 6000.

**Pre-processing of scRNA-sequencing data.** Cell Ranger pipeline (v6.1.2, 10x Genomics)[57] was used to demultiplex samples, process barcodes, and align reads to the GRCh38 reference genome provided by Cell Ranger (v6.1.2). The "mkfastq" module of Cellranger was used to demultiplex raw base call (BCL) data produced by Illumina sequencers into FASTQ files. The "count" module of Cellranger was used for alignment, filtering, barcode, and UMI counting. Single-cell gene count matrices were constructed for each sample using Seurat (v4.3.0)[58] with cells that had number of detected features ≥ 500 and features that had number of detected cells ≥ 5. Given varying distribution of nCount_RNA, nFeature_RNA, and percent.mt among respective samples, we implemented distinct maximum cutoff thresholds (nCount_RNA < 50,000–300,000, nFeature_RNA < 9000–12000, and percent.mt < 5–15). To remove cells considered as doublets, scDblFinder (v1.14.0)[59] detected doublets using the expected doublet rate of 0.8 %.

Single-cell gene count matrices were merged and log-normalized using "NormalizeData" function. 2000 top highly variable genes were selected based on average expression and dispersion for each gene with VST selection methods using the "FindVariableFeatures" function. All respective samples were feature-level scaled using "ScaleData" to reduce the effects of outliers.

To remove batches derived from each sample, "RunFastMNN" was used to integrate Seurat object using batchelor (v1.13.3)[60]. When excitatory neurons were extracted from the object and pre-processed, the anchors between 13 datasets were identified using "FindIntegrationAnchors" and utilized to integrate all datasets through "IntegrateData".

For the analysis of scRNA-seq data of cortical organoids, publicly available single-cell datasets from eleven independent, widely utilized cortical organoids, developed using four different methods in previous studies (Qian, et al.[20], n = 1; Xiang, et al.[8], n = 2; Yoon, et al.[37], n = 3; Kadoshima, et al.[5], n = 5) were obtained from GEO database (GSE137941; GSE98201, GSE107771, and GSE132672, respectively). For organoids developed using methods from Yoon, et al.[37], we specifically utilized the hCS-FF scRNA-seq dataset (n = 3), which comprises cortical organoids derived from hPSCs under feeder-free conditions, among the various datasets included in their study. Additionally, 13 independent cortical organoids from seven datasets included in the HNOCA study (Birey, et al.[4], n = 1; Kelava, et al.[43], n = 3; Khan, et al.[41], n = 2; Pellegrini, et al.[44], n = 1; Sloan, et al.[22], n = 1; Trujillo, et al.[45], n = 1; Vértesy, et al.[42], n = 4) were obtained from the GEO database (GSE93811, GSE187877, GSE145122, GSE150903, GSE99951, GSE130238, and GSE205554, respectively). For cortical organoids developed using methods by Qian, et al.[20], Xiang, et al.[8], Yoon, et al.[37], Birey, et al.[4], Kelava, et al.[43], Khan, et al.[41], Pellegrini, et al.[44], Sloan, et al.[22], Trujillo, et al.[45], or Vértesy, et al.[42], single-cell datasets generated by the original publications were used. For organoids developed using the method by Kadoshima, et al.[5] single-cell datasets generated by Bhaduri, et al.[61] were used. Seurat objects for these datasets were created using features detected in at least 10 cells and cells having at least 500 features. Due to the diverse distribution of percent.mt across different samples, we applied specific maximum cutoff thresholds ranging from 5 % to 15 % for each sample. The remaining processes, including the removal of doublets, pre-processing, and batch correction, were conducted under the same conditions as described above.

**Clustering.** The nearest neighbors present in the single-cell data were found with "FindNeighbors" function using dimensions from NMF. Graph-based Louvain clustering was done with "FindClusters" function. Thereafter, the 50 NMFs were used for t-distributed Stochastic Neighbor Embedding (t-SNE) non-linear dimensionality reduction using RunTSNE.

**Cell-type annotation.** Cell-type annotations of identified clusters of current cortical organoids and cortical assembloids were performed by comparison of marker genes in each cluster to those of previously annotated cell types[61]. The genes used to annotate each cluster are provided in Supplementary Data 1 and 2. When marker-gene based annotation was not available, literature-based annotation was used.

**Annotation of six cortical layers.** Excitatory neuron clusters were re-clustered at a resolution specific to each sample and annotated to one of the six cortical layers according to their correlation coefficient with the six cortical layers in the human brain. Using spatialLIBD (v1.10.1)[62], the correlation between the t-statistics from gene enrichment analysis of histological layers in the reference dataset and the t-statistics from gene enrichment in our query dataset was calculated. The human DLPFC 10x Genomics Visium dataset from Maynard, et al.[63] was used for the analysis. The clusters of our query dataset were annotated, corresponding to the neuronal layers with the highest correlation coefficients. Layer 1 was annotated based on *RELN* expression, and the remaining clusters were assigned to the layers with the highest correlation coefficient scores. Clusters that did not show a positive correlation with any of the six layers in the human brain are designated as "undefined".

**Developmental trajectory analysis.** Partition-based graphical abstraction (PAGA) function, available in the Scanpy library (v1.9.5), within the Python environment (v3.8.18), was employed to create a simplified graph representation of partitions. Utilizing PCA cell embeddings, a neighborhood graph was computed with parameters set to n_pcs = 50

and n_neighbors = 20. Subsequently, the connectivity patterns among partitions were visualized, with edge weights indicating the strength of connections.

**TMAP analysis.** TMAP analysis was performed following the methodology as previously described[64,65]. BrainSpan RNA-seq data[66] was used as the primary human brain reference to compare differential gene expression patterns between the human brain during developmental stages and cortical assembloids. The gene expression patterns of the human brain were derived from "brainSpan_pariedVoom_results.rdata", as utilized in previous studies[64]. In brief, gene expression levels from RNA-seq data of the human brain tissues were normalized and grouped into 13 stages, ranging from 8 PCW to 40 years. Fold change for each developmental stage was calculated by comparing it to the baseline values of the earliest stage in vivo (stage 2, 8–10 PCW) using the limma-voom method[67].

Gene expression values from scRNA-seq data of 13 cortical assembloids were aggregated to generate pseudobulk expression data for each sample. Gene expression values from pseudobulk expression data of 13 cortical assembloids and bulk RNA-seq data of 3 early brain organoids (day 32) were normalized, and fold change was calculated for each of the 13 cortical assembloids by comparing them to the 3 early brain organoids using DESeq2 (v1.40.2)[68]. Genes were then ranked by log fold change (logFC), and the rank-rank hypergeometric test[69] was employed to calculate the significance of the overlap of the genes, using a step size of 200 genes[65].

**Calculation of MI score.** MI score was calculated based on the previous study[24]. MI score was calculated between clusters and individual assembloids with mpmi (v0.43.2.1). The statistical significance of the observed MI scores was calculated by generating background distributions for each dataset.

**Integrative analysis of cortical assembloids and cortical organoids.** scRNA-seq datasets from cortical assembloids and 11 publicly available cortical organoid studies[5,8,20,37] were integrated using the merge function in Seurat. Raw count matrices were normalized with the NormalizeData function (scale factor = 10,000), and the top 2000 highly variable genes (HVGs) shared across datasets were selected with FindVariableFeatures for downstream analysis. The data were scaled using ScaleData, followed by PCA on the HVG matrix with the RunPCA function. To correct for batch effects, we applied RunHarmony with the top 30 principal components, specifying group.by.vars as data type, thereby generating an integrated expression space for comparative analysis. Cell types were annotated for each cluster of cortical organoids and assembloids based on marker gene expression profiles. To analyze six cortical layers, excitatory neuron clusters were re-clustered and manually assigned to layers L1–L6 according to layer-specific marker gene expression.

**Integrative analysis of cortical assembloids and human fetal brains.** scRNA-seq datasets from cortical assembloids and publicly available human fetal brain (cortex) samples from the second and third trimesters[70] were integrated using the anchor-based workflow in Seurat. Datasets were merged and normalized with the LogNormalize method, and highly variable genes (HVGs) were identified using the variance-stabilizing transformation (VST) via FindVariableFeatures. Integration features were selected with SelectIntegrationFeatures, anchors were identified with FindIntegrationAnchors, and datasets were harmonized with IntegrateData to generate a shared integrated expression space. Cell clusters in the integrated object were annotated to canonical cell types based on marker gene expression.

To assess transcriptional similarity between cortical assembloids and the human fetal cortex, Pearson correlation analysis was performed using log-normalized gene expression values (mean expression per gene). For each shared cell type, expression profiles were summarized by averaging gene-level expression within each sample (cortical assembloids, n = 4; human fetal cortex, n = 13 spanning the second and third trimesters). Pairwise Pearson correlation coefficients (PCCs) were then calculated between assembloids and fetal cortex for the same cell type, thereby quantifying transcriptional similarity across developmental stages.

To evaluate cell composition similarity, we compared the relative proportions of cell types between cortical assembloids and the human fetal cortex. For each fetal brain sample (n = 13), Euclidean distances were calculated between its cell-type composition vector and those of the cortical assembloids (n = 4). Distances were then averaged across the four assembloids to yield a single similarity score for each fetal brain sample.

**Statistics and reproducibility**
Statistical analyses were performed using GraphPad Prism ver. 10. Values of $p < 0.05$ were considered statistically significant. No statistical methods were used to predetermine sample size. The experiments were not randomized, and investigators were not blinded to allocation during experiments and outcome assessment.

Figure 1c: (top left) Quantification of the NPC population by counting the number of SOX2-positive cells in each rosette. The number of NPCs in all rosettes was quantified in two sections of one organoid. All organoids in each group were counted. hESC (H9) derived organoids_batch 1 (control, n = 148; CHIR and SAG, n = 74), hESC (H9) derived organoids_batch 2 (control, n = 175; CHIR and SAG, n = 83), hiPSC #1 (IMR90) derived organoids (control, n = 146; CHIR and SAG, n = 102), hiPSC #2 (GM25256) derived organoids (control, n = 136; CHIR and SAG, n = 77), hiPSC #3 (GM23338) derived organoids (control, n = 195; CHIR and SAG, n = 69). Significance was calculated using an unpaired two-tailed t-test. (top right) Quantification of VZ thickness by measuring the thickness of VZ from two independent regions in each rosette (at 90-degree angle intervals). The VZ thickness of all rosettes was quantified in two sections of one organoid. All organoids in each group were counted. hESC (H9) derived organoids_batch 1 (control, n = 296; CHIR and SAG, n = 148), hESC (H9) derived organoids_batch 2 (control, n = 350; CHIR and SAG, n = 166), hiPSC #1 (IMR90) derived organoids (control, n = 292; CHIR and SAG, n = 204), hiPSC #2 (GM25256) derived organoids (control, n = 272; CHIR and SAG, n = 154), hiPSC #3 (GM23338) derived organoids (control, n = 390; CHIR and SAG, n = 138). Significance was calculated using an unpaired two-tailed t-test. (bottom left) Quantification of the rosette size by measuring the diameter of SOX2+ rosette. All rosettes were quantified in two sections of one organoid. All organoids in each group were counted. hESC (H9) derived organoids_batch 1 (control, n = 148; CHIR and SAG, n = 74), hESC (H9) derived organoids_batch 2 (control, n = 175; CHIR and SAG, n = 83), hiPSC #1 (IMR90) derived organoids (control, n = 146; CHIR and SAG, n = 102), hiPSC #2 (GM25256) derived organoids (control, n = 136; CHIR and SAG, n = 77), hiPSC #3 (GM23338) derived organoids (control, n = 195; CHIR and SAG, n = 69). Significance was calculated using an unpaired two-tailed t-test. (bottom right) Quantification of the number of rosettes. The number of rosettes was quantified in three sections of one organoid. All organoids in each group were counted. hESC (H9) derived organoids_batch 1 (control, n = 18; CHIR and SAG, n = 15), hESC (H9) derived organoids_batch 2 (control, n = 21; CHIR and SAG, n = 18), hiPSC #1 (IMR90) derived organoids (control, n = 15; CHIR and SAG, n = 18), hiPSC #2 (GM25256) derived organoids (control, n = 15; CHIR and SAG, n = 15), hiPSC #3 (GM23338) derived organoids (control, n = 18; CHIR and SAG, n = 15). Significance was calculated using an unpaired two-tailed t-test. Fig. 1f: (left) Three biological replicates were evaluated (n = 3). (right) Three biological replicates were evaluated (n = 3). Significance was calculated using an unpaired two-tailed t-test. Figure 1g–k: Representative images shown are from experiments

repeated independently at least three times, with similar results. Figure 1l: (left) Representative images shown are from experiments repeated independently at least three times, with similar results. (right) The size of the assembloids was determined by averaging the diameters of the assembloids, measured twice at a 90-degree angle.

Figure 2a: Representative images shown are from experiments repeated independently at least three times, with similar results. Figure 2c: For layer quantification, the image of cortical plate of cortical assembloids is evenly divided into 15 bins, spanning from apical to basal directions. For each bin, the proportion of layer-specific marker-positive cells is calculated as [number of layer-specific marker-positive cells/number of total neurons]. Each bin is assigned to one of the six cortical layers based on the major cell population comprising each bin, determined by the proportion of layer-specific marker-positive cells. The thickness of six cortical layers was measured in four independent regions of each section (at 90-degree angle intervals), and two sections were quantified for one assembloid (n = 8). Five (for hESC derived assembloids) or three (for hiPSC derived assembloids) biological replicates were evaluated. Figure 2d: Representative images shown are from experiments repeated independently at least three times, with similar results.

Figure 3b: Five sections were quantified for one assembloid (n = 5). Five (for hESC derived assembloids) or three (for hiPSC derived assembloids) biological replicates were evaluated. Fig. 3e: The number of spikes and the amplitude were quantified in five selected cells for one assembloid (n = 5). Five (for hESC derived assembloids) or three (for hiPSC derived assembloids) biological replicates were evaluated. Fig. 3h: The signals from all active electrodes were quantified for each assembloid (n = 1). Five (for hESC derived assembloids) or three (for hiPSC derived assembloids) biological replicates were evaluated.

Figure 4a: Cortical assembloids (n = 13; hESC (H9): five assembloids per batch, two batches analyzed; hiPSC #1 (IMR90), #2 (GM25256), and #3 (GM23338): one assembloid from one batch analyzed), cortical organoids (Qian, et al.[20] n = 1; Yoon, et al.[37] n = 3; Xiang, et al.[8], n = 2; Kadoshima, et al.[5], n = 5).

Figure 5g: Cortical assembloids (n = 7), cortical organoids (Yoon, et al.[37], n = 6; Xiang, et al.[8], n = 5; Kadoshima, et al.[5], n = 5). Significance was calculated using an unpaired two-tailed t-test.

Supplementary Fig. 1c: Three biological replicates were evaluated (n = 3). Significance was calculated using an unpaired two-tailed t-test.

Supplementary Fig. 2e: Membrane resistance (left), capacitance (middle), and resting membrane potential (right) of a cell from cortical assembloids measured using whole-cell patch-clamp recording (ten cells were recorded; n = 10).

Supplementary Fig. 6a: Cortical assembloids (n = 13; hESC (H9): five assembloids per batch, two batches analyzed; hiPSC #1 (IMR90), #2 (GM25256), and #3 (GM23338): one assembloid from one batch analyzed), cortical organoids (Yoon, et al.[37], n = 3; Xiang, et al.[8], n = 2; Kadoshima, et al.[5], n = 5).

Supplementary Fig. 7a: Cortical assembloids (n = 13; hESC (H9): five assembloids per batch, two batches analyzed; hiPSC #1 (IMR90), #2 (GM25256), and #3 (GM23338): one assembloid from one batch analyzed), cortical organoids (Qian, et al.[20], n = 1; Yoon, et al.[37], n = 3; Xiang, et al.[8], n = 2; Kadoshima, et al.[5], n = 5).

Supplementary Fig. 8a, c: Cortical assembloids (n = 4; hESC (H9), hiPSC #1 (IMR90), #2 (GM25256), and #3 (GM23338): one assembloid from one batch analyzed), cortical organoids (Birey, et al.[4], n = 1; Kelava, et al.[43], n = 3; Khan, et al.[41], n = 2; Pellegrini, et al.[44], n = 1; Sloan, et al.[22], n = 1; Trujillo, et al.[45], n = 1; Vértesy, et al.[42], n = 4). Supplementary Fig. 8g: Cortical assembloids (n = 7), cortical organoids (Kelava, et al.[43], n = 4; Khan, et al.[41], n = 4; Vértesy, et al.[42], n = 6). Significance was calculated using an unpaired two-tailed t-test.

Supplementary Fig. 9a: Cortical organoids (Kelava, et al.[43] n = 3; Khan, et al.[41], n = 2; Vértesy, et al.[42], n = 4).

Supplementary Fig. 10a: Cortical organoids (Birey, et al.[4], n = 1; Kelava, et al.[43], n = 3; Khan, et al.[41], n = 2; Pellegrini, et al.[44], n = 1; Sloan, et al.[22], n = 1; Trujillo, et al.[45], n = 1; Vértesy, et al.[42], n = 4).

Supplementary Fig. 11a, f: Cortical assembloids (n = 4; hESC (H9), hiPSC #1 (IMR90), #2 (GM25256), and #3 (GM23338): one assembloid from one batch analyzed), cortical organoids (Qian, et al.[20], n = 1; Yoon, et al.[37], n = 3; Xiang, et al.[8], n = 2; Kadoshima, et al.[5], n = 5; Birey, et al.[4], n = 1; Kelava, et al.[43], n = 3; Khan, et al.[41], n = 2; Pellegrini, et al.[44], n = 1; Sloan, et al.[22], n = 1; Trujillo, et al.[45], n = 1; Vértesy, et al.[42], n = 4). Supplementary Fig. 11i: Cortical assembloids (n = 4; hESC (H9), hiPSC #1 (IMR90), #2 (GM25256), and #3 (GM23338): one assembloid from one batch analyzed).

### Reporting summary
Further information on research design is available in the Nature Portfolio Reporting Summary linked to this article.

## Data availability
The scRNA-seq data of cortical assembloids generated in this study have been deposited in the Korea BioData System (K-BDS) database under accession code KAP241776. Other data generated in this study are provided in the Supplementary Information and Source Data files. The processed scRNA-seq data of the human fetal brain cortex were obtained from the following link: [https://cellxgene.cziscience.com/collections/ad2149fc-19c5-41de-8cfe-44710fbada73]. The scRNA-seq data of previously published cortical organoids from Birey, et al.[4], Kadoshima, et al.[5], Kelava, et al.[43], Khan, et al.[41], Pellegrini, et al.[44], Qian, et al.[20], Sloan, et al.[22], Trujillo, et al.[45], Vértesy, et al.[42], Xiang, et al.[8], and Yoon, et al.[37] were obtained from the GEO database under the following accession codes, respectively:

GSE93811 [https://www.ncbi.nlm.nih.gov/geo/query/acc.cgi?acc=GSE93811],

GSE132672 [https://www.ncbi.nlm.nih.gov/geo/query/acc.cgi?acc=GSE132672],

GSE187877 [https://www.ncbi.nlm.nih.gov/geo/query/acc.cgi?acc=GSE187877],

GSE145122 [https://www.ncbi.nlm.nih.gov/geo/query/acc.cgi?acc=GSE145122],

GSE150903 [https://www.ncbi.nlm.nih.gov/geo/query/acc.cgi?acc=GSE150903],

GSE137941 [https://www.ncbi.nlm.nih.gov/geo/query/acc.cgi?acc=GSE137941],

GSE99951 [https://www.ncbi.nlm.nih.gov/geo/query/acc.cgi?acc=GSE99951],

GSE130238 [https://www.ncbi.nlm.nih.gov/geo/query/acc.cgi?acc=GSE130238],

GSE205554 [https://www.ncbi.nlm.nih.gov/geo/query/acc.cgi?acc=GSE205554],

GSE98201 [https://www.ncbi.nlm.nih.gov/geo/query/acc.cgi?acc=GSE98201],

GSE107771 [https://www.ncbi.nlm.nih.gov/geo/query/acc.cgi?acc=GSE107771]. Source data are provided with this paper.

## Code availability
All analyses were performed using publicly available packages, tools, and established codes, including Seurat (https://satijalab.org/seurat/), Cell Ranger (https://share.google/ouaP1k3vKXVRJ9Ty0), spatialLIBD (https://www.bioconductor.org/packages/release/data/experiment/html/spatialLIBD.html), Scanpy (https://share.google/3t5KpWj0CXIaW3zG4), DESeq2 (https://bioconductor.org/packages/release/bioc/html/DESeq2.html), and mpmi (https://cran.r-project.org/web/packages/mpmi/index.html). Details of the analysis workflow, parameters, and functions used are described in the "scRNA sequencing" section of the "Methods".

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

## Acknowledgements

We thank Mi-Ryoung Song at Gwangju Institute of Science and Technology for their generous provision of the RELN-expressing construct. This research was supported by grants from the National Research Foundation of Korea RS-2023-00223277 (K.S.), RS-2024-00466703 (K.S.), RS-2022-NR072198 (E.K.), Samsung Science and Technology Foundation SSTF-BA2101-12 (K.S.), New Faculty Startup Fund from Seoul National University (K.S.), and the BK21FOUR Research Fellowship (K.S.).

## Author contributions

E.K., Y.K., and K.S. conceived the ideas and experimental design. E.K., Y.K., S.H., I.K., and J.L. performed the overall experiments. Y.K., J.K., K.Y., H.L., and J.C. analyzed scRNA-seq data. J.Y. performed whole-cell patch-clamp experiments, and J.H.K. helped analyze data from electrophysiological experiments. E.K., Y.K., and K.S. wrote the manuscript.

## Competing interests

The authors declare no competing interests.
