## [Transparent Peer Review file · Nature Communications]

Single rosette-based generation of uniform cortical assembloids recapitulating cellular interactions between neurons and glial cells

Corresponding Author: Dr Kunyoo Shin

Version 0:

Reviewer comments:

Reviewer #1

(Remarks to the Author)

I appreciate the revision and the additional experiments/analysis by Kim et al. However, some of the answers to my previously raised concerns have not been satisfactorily addressed and I suggest further rephrasing of some of the statements. I believe that the manuscript presents interesting results, and would benefit from a more balanced perspective.

I have previously requested the authors to somehow soften their claims regarding the low reproducibility of previously published brain organoids protocols, especially since in my opinion problems of reproducibility are evident also in the current manuscript (e.g. Fig 1d and 2c-d). I appreciate that the authors now incorporated a table (supp 1) in which they summarized multiple findings from previous studies, but I find some of the evaluations subjective and not nuanced (e.g. heterogeneity is described as “high” for all the previous protocols except 1 or NA). What is the assessment criteria? This should be revised.

Concerning my scalability concern. I appreciate the inclusion of a step-by-step protocol and additional details, but the generation of these organoids remains very laborious, and this should clearly mentioned.

Regarding the single cell sequencing analysis, as showed in Figure 4-5 and relative supplement, I have requested to show data integration. I think it can help to better compare the different data sets, and I have requested this type of analysis also for the human fetal brain tissue. Single cell sequencing data sets are available for human cortex of different age, and it would place the study in better context to integrate the data.

Reviewer #2

(Remarks to the Author)

In response to my comments, the authors have made numerous changes to the manuscript. Unfortunately, however, many of the statements they made before have remained and I continue to disagree with many of the conclusions made in this paper. This is very unfortunate, because the method described in this paper per se is interesting and worth publishing. But some of the statements just can not be left uncommented.

Below, I re-state my most important previous comments and respond to the author's rebuttal. I would like to emphasize that my comments are meant to be constructive, paving a way forward to publishing.

Comment 1: The authors claim that current organoid protocols are limited by “lack of sustained proliferation” and by “... absent populations of late born neurons”, etc. They cite a number of papers published eight years ago. However, there are numerous publications (particularly single-cell analysis) revealing that outer layer neurons are indeed generated in organoids (for example: Quadrato, et al., 2017 (called callosal neurons here), . Furthermore, depending on culture

conditions, organoids can also sustain proliferation for long times. The authors should be more cautious in stating the problem they aim at overcoming.

Response and Assessment 1: In response to the comments above, the authors conducted an extensive review of the current literature, outlining both the limitations and capabilities of existing organoid protocols across key parameters: structural organization, functional maturity, cellular diversity, and overall heterogeneity. While this is an impressive effort, I am concerned that “grading” all previous research in the field would belong into a review article and does not fit into a research article. This is particularly true because the author’s assessment appear to be subjective, without explanation of standardized quantification systems that were applied. Furthermore, the authors fail to apply the same grading system to their own cortical assembloids, significantly weakening their claims.

Furthermore, there are numerous points throughout the paper where the authors make statements that I disagree with. For example, “In addition, most current cortical organoids are composed of deep-layer neurons, with a relatively lower proportion of upper layer neurons”. There are multiple examples of organoids allowed to mature sufficiently so that they generate upper layer neurons, which arise later in development^{1–3}. Furthermore, the subjective interpretation of previous publications by the authors raises other concerns. For example, the authors do not mention that a previous study, Qian et al., 2020, has demonstrated layer formation of reelin positive cells using a different protocol⁴. This study is described in their table, but is not addressed in the text which fails to adequately establish that cortical layering has been achieved to various degrees by other groups.

Therefore, while the author’s response to earlier comments provides some insight into the field, it does not adequately address the core of the issue. While I agree that organoid technology has limitations and should be used appropriately and further developed, the authors overstate these limitations in what seems like an effort to provide a more dramatic background by which their findings can be contrasted. This is a key weakness of the manuscript and has not been appropriately addressed.

Comment 2: It will be important for the authors to determine the dorso-ventral identity of the organoids and the neurons contained in them. The protocol uses a combination of both Shh and Wnt signalling activation, which provides both dorsalizing and ventralizing cues. Therefore, it will be important to demonstrate dorso-ventral identity both at the progenitor and at the neuron stage. Single-cell analysis reveals only minor numbers of interneurons. This could be problematic as pure assemblies of excitatory neurons are not expected to recapitulate neural network formation. Better characterization of the origin and distribution of the interneurons will be important. Are there “islands” of ventral progenitors forming in the organoids?

Response and Assessment 2: In response to the comments above, the authors performed immunostaining and gene expression analyses to identify the lineage of the organoids generated using their method (Figure R1). Their analysis concludes that the tissue is predominantly dorsally patterned, with cells positive for PAX6 and TBR1. The authors further used immunostaining to identify the potential lineage of the limited inhibitory interneurons present, attributing these to sporadic DLX1-positive cells (Figure R4). There remain, however, multiple points which remain problematic. These are listed below:

1. Assessing the lineage commitment of progenitors in their system is critical and rather than providing these data solely in revision responses and not in the main manuscript, I encourage the authors to incorporate their data into the main manuscript.

2. The broader issue of using CHIR and SAG (Wnt and Hedgehog agonists, respectively) as mitogens while disregarding their morphogenic effects remains confusing. These compounds are well established as having a dramatic effect on lineage commitment of neuronal progenitors and are essentially ignored by the authors without strong rationale. The rationale presented—that their use at early stages allows for primarily mitogenic activity—is noted; however, compared to other protocols, days 25–32 is not considered early within typical patterning timelines⁵. While the end point seems to be that their tissue is predominantly dorsally patterned, this significant deviation from established systems is poorly addressed. For instance, if we accept the conclusions from the immunostaining, how do the authors explain the apparent lack of morphogenic effect?

3. The authors use well established markers for excitatory lineages (i.e., dorsal (PAX6 and TBR1) but only use DLX1 as validation for the lack of inhibitory neuron progenitors for which there are numerous, region specific, markers. In order to make this conclusion stronger, the authors could broaden their scope of ventral markers to include more specific identifiers of inhibitory neuron lineages, many of which have been applied already in organoids. These may include NKX2-1 (Medial Ganglionic Eminence), GSX2, LHX6 (Lateral Ganglionic Eminence), SCGN and COUP-TFII (Caudal Ganglionic Eminence), DLX2, SOX6 (migrating GABAergic progenitors), many of which have been used in other studies. Taken together, the authors have assessed the developmental lineage of their cortical assembloids to a degree and have partially addressed earlier comments. However some broader conceptual concerns remain.

Comment 3: The radial organization of progenitors and growing axons is impressive but needs to be supported by statistical analysis. Labelling of individual radial glia cells and individual neurons will be important to make this point. This can easily be done, for example using an adenovirus

Response and Assessment 3: The authors have addressed this point by using a combination of viral vectors expressing ubiquitous EGFP and SOX2 immunostaining. The rationale is that SOX2/EGFP co-positive cells represent radial glial populations, while SOX2-negative, EGFP-positive cells represent neurons. By quantifying the number of SOX2+ cells oriented within $\pm 30^\circ$ of a line drawn perpendicular to the tangential surface of the ventricular zone, they were able to identify radially organized cells.

Overall, this approach addresses my earlier comments.

Comment 4: The authors use the term “assembloid” to describe their cultures. I am aware that this is in agreement with Pasca et al., Nature 2022. On the other hand, a related technology (Antón-Bolaños, et al., Nature 2024) has recently been called a “chimeroid”. I am afraid that this difference in nomenclature might be slightly confusing and would suggest the authors to add a clarifying sentence (for example, explaining why they chose assembloids over chimeroid).

Response and Assessment 4: In response to my earlier comments, the authors have provided a rationale for their choice of “assembloid” versus “chimeroid” nomenclature and therefore addressed my earlier concern.

Comment 5: The consistency of the MEA signals is impressive. But it might be misleading: Besides excitatory neurons, assembly of functional circuits requires inhibitory interneurons, which are not present in the cultures. In my view, this is a major weakness of the method that should be mentioned in the text.

Response and Assessment 5: The authors have partially addressed this concern by noting the absence of inhibitory interneurons in the discussion section. However, it remains unclear how this limitation is connected to the electrophysiological activity presented in Figure 3. While the authors point out that other studies recapitulating dorsal forebrain development similarly lack interneurons, they have not directly discussed how this deficiency might affect the functionality of their cortical assembloids.

This point should be addressed more explicitly. Although Figure 3 demonstrates that the cortical assembloids exhibit a degree of functionality, the authors have not compared their results to other models that either include or exclude inhibitory neuron populations. This remains a significant weakness, as it is unclear whether the single-rosette method improves tissue functionality relative to existing approaches. In order to make this conclusion stronger, the authors should demonstrate, preferably experimentally, that the electrophysiological activity displayed here is on par or an improvement when compared to other similar studies which lack interneurons.

Response 6

This concern has not been fully addressed. While the author’s overall conclusion that cortical assembloids display consistent properties appears valid and is an important contribution to the field, there are significant issues with how the analyses comparing their method to previously published protocols were conducted, which weakens the impact of their claims. Some major points are listed below:

1. The analysis presented in Figures 4C and 5C appears inappropriate for comparing cell type-specific and layer marker expression across protocols. The authors use different marker genes across datasets, preventing a direct and fair comparison. If the authors are using specific marker genes in their own protocol, the same markers should be applied across datasets to properly assess variability, rather than selecting different genes from different papers.
2. The authors claim that neurons in their cortical assembloids segregate into six distinct populations expressing layer-specific markers. However, this same degree of specification appears present in other datasets as well (e.g., Supplementary Figure 7), albeit in lower amounts. Therefore, the authors should limit their claims to their data, that demonstrates they can achieve robust amounts of the separate layers, but not necessarily their existence, as other studies can also achieve this.
3. The stacked bar plots in Figure 4B do not convincingly support the claim that their method significantly reduces the percentage of radial glial cells compared to other protocols as often times their cortical assembloids can consist of over 30% radial glial cells, similar to other studies. While reduced batch variability seems to be the authors' strongest argument, the assertion that other methods consistently have higher radial glial proportions than theirs is not strongly supported by the data presented.
4. The authors argue that other protocols suffer from inconsistent interneuron production. But this critique applies equally to their own system, where inhibitory neurons are also largely absent. It is therefore unclear to me how their method improves upon this.
5. The authors note the absence of microglia in other protocols. However, it is well-established that microglia must typically be added exogenously to brain organoid cultures. This makes the comparison somewhat unfair. If the authors wish to strengthen their point regarding microglia, they should compare their findings to protocols where microglia were intentionally incorporated into organoids^{6,7} and assess survival, integration, and transcriptional fidelity (referenced in Line 288 - 289).

References

1. Quadrato, G. et al. Cell diversity and network dynamics in photosensitive human brain organoids. *Nature* 545, 48–53 (2017).
2. Uzquiano, A. et al. Proper acquisition of cell class identity in organoids allows definition of fate specification programs of the human cerebral cortex. *Cell* 185, 3770-3788.e27 (2022).
3. He, Z. et al. An integrated transcriptomic cell atlas of human neural organoids. *Nature* 635, 690–698 (2024).
4. Qian, X. et al. Sliced Human Cortical Organoids for Modeling Distinct Cortical Layer Formation. *Cell Stem Cell* 26, 766-781.e9 (2020).
5. Pasca, A. M. et al. Functional cortical neurons and astrocytes from human pluripotent stem cells in 3D culture. *Nature Methods* 12, 671–678 (2015).
6. Cakir, B., Kiral, F. R. & Park, I.-H. Advanced in vitro models: Microglia in action. *Neuron* 110, 3444–3457 (2022).
7. Popova, G. et al. Human microglia states are conserved across experimental models and regulate neural stem cell responses in chimeric organoids. *Cell Stem Cell* S1934590921003763 (2021) doi:10.1016/j.stem.2021.08.015.

Reviewer #3

(Remarks to the Author)

The work claims that the authors can reconstitute various cell modalities to form a highly complex, mature, homogeneous six-layered cortex, termed cortical assembloids. Showing six-layered cortices that develop from a germinal layer of neurogenesis with functionality is a significant advance, which is highly appreciated in the field. The authors use the term reconstitution, and they claimed that they could reconstitute various cellular compositions to form a structurally brain-like tissue system with six layers of cortex. Conceptually, it is a fascinating idea, even though the layered cortex does not emerge from the typical ventricular zone, and the data are of high quality.

By definition, reconstitution means to break down a biochemical complex or a tissue and put it back in a functional form. This is where the concept is not solid because reconstitution is not reflected in the work. The work involves a scaffold of neural stem cells (Sox2-positive cells) and is then supplemented with reelin-positive neurons, which surprisingly and nicely guide the six layers. This is not a reconstitution, but rather driven by a sequential differentiation process, which is equally exciting. Naming it and extensively discussing the process of cellular reconstitution is misleading and incorrect. My further comments are provided below, and I hope they will be helpful to the authors in restructuring the work into an acceptable form. Data quality is, of course, of high quality, but it must be carefully analyzed to its depth to derive a solid conclusion.

The authors must avoid strong statements without any basis, which already appear in the abstract, that "Current brain organoid technology fails to provide adequate patterning cues to induce a mature structure that represents the complexity of the human brain." Which of the ones fail to show the patterning? I could name dozens of them who have dramatically developed the methodology. I found their statement to be blunt and somewhat unfair. The presented work does not show any early patterning event (see below)

"Modular-based cellular reconstitution technology" is indeed a nice term to read, and the manuscript must explain the logic behind it. The power of the newly developed method can be witnessed only when it is used to model a disease. Otherwise, the authors' claims remain questionable. Without presenting one or two applications, the work is merely descriptive of the current state of brain organoid technology and thus falls short of publication here.

The authors conducted an extensive literature review, covering over 60 papers, and presented a scoring system indicating that all fall short in some way. It seems like a gross overstatement, and I question the logic behind the scoring system. They cite reference 23, which has no relevance to brain organoids. I am unsure of its origin, other than that it is their own citation. The modular assembly shown in their work is very challenging and complex, and I appreciate it as the authors have tried it with at least three iPS donors and an ES cell line. The scheme illustrates the general method, but the data lacks details. How were the organoids dissected manually to get one rosette containing tissue, and how is it possible to do this in large numbers without any variations? The method also does not control the number of pluripotent cells used (says 2-3 clones or something). SAG and CHIR (agonists of growth signalling) opposes each other. For example, elevated SHH signaling in vivo prevents the neural tube from closing, which explains why the authors observe a large rosette-like structure. Then, what was the logic in using CHIR in addition to SAG? How could the authors count SOX2-positive cells in a rosette of a 3D tissue?

I don't question the technical soundness of the method, but it lacks content and details. It is, by far, a very complex technology to adapt to reproduce and simultaneously minimize intra- and inter-batch differences. With this complexity, it is challenging to generate a large number of organoids, and it is unarguably understandable.

Other fundamental problems here are the apicobasal polarity, which must be demonstrated. I don't see any indication for neurons in Shh and CHIR-treated organoids (before they seed reelin cells), and they are just too big and don't look like there is a clear apicobasal polarity. In fact, for the first few days, a method to generate forebrain organoids has been adapted. How come there are no neurons at the basal side? From this, I seriously suspect that the Sox2-positive structures are merely unphysiological, and this must be tested for the apicobasal markers, such as ZO1 and the pan-neuronal marker TUJ1. I guess that the shown are a cluster of SOX2-positive cells in a perturbed, enlarged, and broken neural rosette. In the displayed images, I do see two or three rosette-like structures of two different sizes (in contrast to what the authors claim). Again, the authors have provided other cell types, which were cultured separately, a notable achievement, but one that may not be directly related to embryonic brain developmental cues. This is already obvious as their organoid does not generate inhibitory neurons. A control experiment should be shown in which the Sox2-positive rosette develops without supplementing reelin cells.

Six layers of neurons are the most exciting part of the work. I am curious how the authors could combine 6 markers in one section. Or are these adjacent sections? Then, how do they create a nice layer over 24 μm ? It would be critical to show more than one specific markers for each layer.

Connect figure 2a to figure 1. The rest of the analysis is basically a consequence of those 6-layered neurons, which have been well performed.

Sc-RNA sequencing has been performed and mainly used to compare the cell proportions, which is not very well exploited (again, figures lack units, and as a result, I don't know what those bars are. Secondly, unsupervised clustering of cell types was not shown). In normal brain development, the excitatory neurons emerge from the dorsal telencephalon, and the inhibitory neurons are from the ventral telencephalon (in this work, inhibitory neurons are low). The data shows some sort of trajectory analysis, but the text does not provide the details. Essentially, the authors describe it as the "trajectory follows normal brain development," but what constitutes normal here? There are several issues, and one such issue is injecting microglia in 5 different places, and this is quite an invasive method and will induce injuries to the organoids. Why not spike

microglia and test if they are taken up and integrated? The authors should consider softening the discussion section when comparing their work with other published results that have modeled one or more diseases, demonstrating that the tissues are functional. The current work lacks any application.

Overall, starting with the Sox2 scaffold, the addition of reelin cells forming a sphere with functional neurons is interesting work that requires restructuring, careful analysis, and avoiding overstating statements will develop valuable work.

Version 1:

Reviewer comments:

Reviewer #1

(Remarks to the Author)

I appreciate the novel single-cell sequencing analysis presented in this work, particularly the dataset integration shown in Supplementary Figure 11, which I consider essential. Regarding the main conclusions of the manuscript, I acknowledge that the authors have made satisfactory adjustments in the Introduction to moderate claims about the limitations of previous models. However, I suggest that a similar textual revision be applied to the Discussion as well, and after that I can recommend publication of the manuscript.

Reviewer #2

(Remarks to the Author)

I would like to congratulate the authors on their thorough and thoughtful revisions. The manuscript has improved considerably, and it is clear that the authors have carefully engaged with the comments provided in earlier rounds of review. Across the different points raised, they have added new data, clarified their methodology and rationale, revised language, and incorporated additional references where appropriate. These efforts have significantly strengthened the balance and rigor of the work, and the authors have addressed my previous concerns.

Comment 1 – Literature review and grading

The authors clarified their evaluation framework and revised overstatements while adding key references. These changes improve clarity and balance, and I consider my concern addressed.

Comment 2 – Dorso-ventral identity

The authors explained the timing of CHIR/SAG treatment and expanded analysis with additional ventral markers. Their revisions address my concerns.

Comment 3 – Radial organization

Quantification of radial alignment using SOX2/EGFP co-labeling provides adequate support. This resolves my earlier concern.

Comment 4 – Nomenclature

The authors justified their continued use of “assembloid” over “chimeroid.” This clarification resolves the issue.

Comment 5 – Interneuron deficiency and MEA activity

Comparative MEA recordings and added data address my concern.

Comment 6 – Comparative analyses

The authors acknowledged marker inconsistencies and performed an integrative analysis with 11 datasets using a unified marker set, supporting improved consistency and maturation. They also revised overstatements on cortical layering, radial glia proportions, interneurons, and clarified microglia integration with new comparative analyses. Together these revisions resolve my concerns.

Reviewer #3

(Remarks to the Author)

The revised manuscript contains extensive information, and the author's response is too lengthy, making it difficult to follow my original questions. For example, I did not ask about the functions of CHIR and SHH, but rather about the logic behind them. However, the authors have started explaining a function I am already familiar with. As the second reviewer also pointed out, the logic of using these two different cell lineage factors, which have opposing functions, is very confusing. The authors mention that they used them as a mitogenic factor to stimulate the proliferation of SOX2 cells, citing Nature Neuroscience 19, 888-896 (2016).

This work is irrelevant here. This work demonstrates that constitutive activation of Shh signalling elevates the TBR2-positive cells and eventually drives cortical folding in rodents, whereas the authors used SAG in this study. Note, it is just a ligand. Even if this activates Shh signaling at a level significantly higher than the physiological level, the authors also use Chir, which opposes Shh signaling by activating Wnt. It does not fit and is not physiological. To elucidate this, one must check if TBR2-positive cells are increased in their organoids with and without the addition of SAG and Chir. The unphysiological nature is again demonstrated by the fact that the authors observe a vast number of excitatory neurons with a low number of inhibitory neurons, and they acknowledge this. Elevated Shh should cause ventralization, not dorsal specification, which is a well-established phenomenon.

I appreciate the figure (R4). I notice a loss of apico-basal polarity. There are many TUJ1-positive regions at the apical side, which are again co-localized by the ZO1 staining at the apical side. Overall, the logical sequence of using these two agents

at a specific time point is mostly missing.

I have raised concerns about the analysis of single-cell data. The authors did not re-analyze, but instead explained the trajectory analysis by text editing. This is not possible without analysis.

I remain skeptical about using the term 'reconstitution,' as it is the wrong choice. The authors then defend their position by referencing their earlier work. I don't see 'reconstitution' there either. I have already proposed a specific alternative for using it when referring to the SOX2-positive cell scaffold and reelin-positive cells as guiding cues. The use of the term "modular assembly" makes much more sense.

Regarding disease modeling or its application, the authors have cited their work on BioArXiv. This is beyond my scope, even before assessing the current work, where the suitability of the model system should be examined. However, the authors have already used the current model system to simulate a disease that may not display the physiological nature.

I also appreciate that the authors have acknowledged the complexity of the model and have helped clarify some of the puzzles. For example, the SOX2-positive cells are from slices, not from 3D rosettes. Similarly, many issues still require careful examination and resolution.

Overall, I have serious doubts about the idea that the shown model consistently reflects any aspect of human brain development. This needs to be addressed, and the model should then be validated through applications such as a disease model or perturbation analysis.

Point-by-point response to the reviewers' comments

Reviewer #1 (Remarks to the Author):

I appreciate the revision and the additional experiments/analysis by Kim et al. However, some of the answers to my previously raised concerns have not been satisfactorily addressed and I suggest further rephrasing of some of the statements. I believe that the manuscript presents interesting results, and would benefit from a more balanced perspective.

I have previously requested the authors to somehow soften their claims regarding the low reproducibility of previously published brain organoids protocols, especially since in my opinion problems of reproducibility are evident also in the current manuscript (e.g. Fig 1d and 2c-d). I appreciate that the authors now incorporated a table (supp 1) in which they summarized multiple findings from previous studies, but I find some of the evaluations subjective and not nuanced (e.g. heterogeneity is described as “high” for all the previous protocols except 1 or NA). What is the assessment criteria? This should be revised.

Regarding the reviewer's question about our assessment criteria, we evaluated each study across four primary parameters—structural organization, functional maturity, cellular diversity, and overall heterogeneity. For each parameter, we defined multiple subcategories and conducted a comprehensive review of all available data (including text, figures, and supplementary materials) from each publication to score each subcategory. For heterogeneity, a score of 'NA' was assigned when no experimental validation existed for a given subcategory. Even when data were presented, we still assigned 'NA' if only a single representative image was shown. When multiple organoids were evaluated, the subcategory was scored as 'Low' if the study included representative images and quantitative analyses across multiple organoids that demonstrated clear consistency or a pronounced lack of heterogeneity among replicates; otherwise, it was scored as 'High'. Subcategory scores were then summed to obtain an overall parameter score, which was classified as high, mid, or low based on predefined thresholds. The detailed scoring criteria are provided in Supplementary Data 2 of the original manuscript.

We acknowledge that, despite our systematic evaluation, some subjectivity may remain. In response to the reviewer's comments, we decided to retain these comprehensive reviews (Supplementary Table 1; Supplementary Data 1 and 2) solely as reference materials for the reviewers during revision and have removed them from the main text. We would be, of course, happy to reincorporate this information into the manuscript if the reviewer believes it would be beneficial.

Concerning my scalability concern. I appreciate the inclusion of a step-by-step protocol and additional details, but the generation of these organoids remains very laborious, and this should clearly mentioned.

As suggested by the reviewer, we have revised the text in the Discussion section of our manuscript to describe the limitations of our methodology, particularly the limited scalability of cortical assembloid generation (page 19).

Regarding the single cell sequencing analysis, as showed in Figure 4-5 and relative supplement, I have requested to show data integration. I think it can help to better compare the different data sets, and I have requested this type of analysis also for the human fetal brain tissue. Single cell sequencing data sets are available for human cortex of different age, and it would place the study in better context to integrate the data.

Regarding the comparative scRNA-seq analysis, integration of our in-house cortical assembloid data with multiple public datasets can result in overcorrection of batch effects, which may obscure biological variation or artificially align distinct cellular populations across different datasets. Therefore, we employed a more appropriate pairwise approach, comparing each dataset individually to ensure accurate batch-effect correction.

In the revised manuscript, as suggested by the reviewer, we expanded our comparative scRNA-seq analysis by integrating our cortical assembloid dataset with datasets from eleven recently published cortical organoid models as well as the developing human brain. These integrated analyses further confirm that our assembloids exhibit a high degree of consistency and maturation (Supplementary Fig. 11a-h; Supplementary Data 1), aligning with findings from the analyses of individual datasets (Figs. 4, 5 and Supplementary Figs. 6, 7), and closely resemble the second to third trimester human fetal cortex in cell-type composition and transcriptomic profiles (Supplementary Fig. 11i-m; Supplementary Data 2).

Reviewer #2 (Remarks to the Author):

In response to my comments, the authors have made numerous changes to the manuscript. Unfortunately, however, many of the statements they made before have remained and I continue to disagree with many of the conclusions made in this paper. This is very unfortunate, because the method described in this paper per se is interesting and worth publishing. But some of the statements just can not be left uncommented. Below, I re-state my most important previous comments and respond to the author's rebuttal. I would like to emphasize that my comments are meant to be constructive, paving a way forward to publishing.

We appreciate the reviewer's evaluation of our initial response and their constructive feedback, which has helped improve the clarity of the manuscript and bring it closer to publication.

Comment 1: The authors claim that current organoid protocols are limited by "lack of sustained proliferation" and by "...absent populations of late born neurons", etc. They cite a number of papers published eight years ago. However, there are numerous publications (particularly single-cell analysis) revealing that outer layer neurons are indeed generated in organoids (for example: Quadrato, et al., 2017 (called callosal neurons here), . Furthermore, depending on culture conditions, organoids can also sustain proliferation for long times. The authors should be more cautious in stating the problem they aim at overcoming.

Response and Assessment 1: In response to the comments above, the authors conducted an extensive review of the current literature, outlining both the limitations and capabilities of existing organoid protocols across key parameters: structural organization, functional maturity, cellular diversity, and overall heterogeneity. While this is an impressive effort, I am concerned that "grading" all previous research in the field would belong into a review article and does not fit into a research article. This is particularly true because the author's assessment appear to be subjective, without explanation of standardized quantification systems that were applied. Furthermore, the authors fail to apply the same grading system to their own cortical assembloids, significantly weakening their claims.

Furthermore, there are numerous points throughout the paper where the authors make statements that I disagree with. For example, "In addition, most current cortical organoids are composed of deep-layer neurons, with a relatively lower proportion of upper layer neurons". There are multiple examples of organoids allowed to mature sufficiently so that they generate upper layer neurons, which arise later in development¹⁻³. Furthermore, the subjective interpretation of previous publications by the authors raises other concerns. For example, the authors do not mention that a previous study, Qian et al., 2020, has demonstrated layer formation of reelin positive cells using a different protocol⁴. This study is described in their table, but is not addressed in the text which fails to adequately establish that cortical layering has been achieved to various degrees by other groups.

Therefore, while the author's response to earlier comments provides some insight into the field, it does not adequately address the core of the issue. While I agrees that organoid technology has limitations and should be used appropriately and further developed, the authors overstate these limitations in what seems like an effort to provide a more dramatic background by which their findings can be contrasted. This is a key weakness of the manuscript and has not been appropriately addressed.

We appreciate the reviewer's acknowledgment of our efforts to provide a comprehensive review of brain organoid research. Regarding the reviewer's comment about our assessment criteria, we evaluated each study across four primary parameters—structural organization, functional maturity, cellular diversity, and overall heterogeneity. For each parameter, we defined multiple subcategories and conducted a comprehensive review of all available data (including text, figures, and supplementary materials) from each publication to score each subcategory. For example, in the evaluation of heterogeneity, a score of 'NA' was assigned when no experimental validation was provided for a given subcategory. Even when data were presented, we still assigned 'NA' if only a single representative image was shown. When multiple organoids were evaluated, the subcategory was scored as 'Low' if the study included representative

images and quantitative analyses across multiple organoids that demonstrated clear consistency or a pronounced lack of heterogeneity among replicates; otherwise, it was scored as 'High'. Subcategory scores were then summed to obtain an overall parameter score, which was classified as high, mid, or low based on predefined thresholds. The detailed scoring criteria are provided in Supplementary Data 2 of the original manuscript.

We acknowledge that, despite our systematic evaluation, some subjectivity may remain. In response to the reviewer's comments, we decided to retain these comprehensive reviews (Supplementary Table 1; Supplementary Data 1 and 2) solely as reference materials for the reviewers during revision and have removed them from the main text. We would be, of course, happy to reincorporate it into the manuscript if the reviewer believes it would be beneficial.

Regarding our statement on previous publications concerning cortical layer composition, we agree with the reviewer's comments. To more accurately reflect prior studies, we have softened our original statement and rephrased the relevant text to better describe the cortical layer generation achieved in current organoid models (page 5). We have also cited the studies suggested by the reviewer (refs 2,25,26 in our revised manuscript).

Regarding the comment about layer formation and RELN expression in other studies, we acknowledge the work by Qian et al. (2016 and 2020), which reported RELN expression in their cortical organoids. In these studies, RELN expression begins relatively late (around day 28) and subsequently decreases by day 84. Notably, RELN expression is localized primarily to the edges of multi-rosette organoids, resulting in uneven distribution across individual rosettes. We also acknowledge that these studies reported the presence of cortical layer-like structures; however, the proper organization of cortical layers in the correct spatial order appears to be lacking. In agreement with the reviewer's comment, we have included a more precise description of Qian et al.'s findings on RELN expression and cortical layer formation in the revised manuscript (page 5).

Overall, we have softened and modified our descriptions of the limitations of current cortical organoid technologies in the revised manuscript. If the reviewer remains unsatisfied or finds any statements still insufficiently balanced, we would greatly appreciate specific suggestions and would be happy to make further modifications accordingly.

Comment 2: It will be important for the authors to determine the dorso-ventral identity of the organoids and the neurons contained in them. The protocol uses a combination of both Shh and Wnt signalling activation, which provides both dorsalizing and ventralizing cues. Therefore, it will be important to demonstrate dorso-ventral identity both at the progenitor and at the neuron stage. Single-cell analysis reveals only minor numbers of interneurons. This could be problematic as pure assemblies of excitatory neurons are not expected to recapitulate neural network formation. Better characterization of the origin and distribution of the interneurons will be important. Are there "islands" of ventral progenitors forming in the organoids?

Response and Assessment 2: In response to the comments above, the authors performed immunostaining and gene expression analyses to identify the lineage of the organoids generated using their method (Figure R1). Their analysis concludes that the tissue is predominantly dorsally patterned, with cells positive for PAX6 and TBR1. The authors further used immunostaining to identify the potential lineage of the limited inhibitory interneurons present, attributing these to sporadic DLX1-positive cells (Figure R4). There remain, however, multiple points which remain problematic. These are listed below:

1. Assessing the lineage commitment of progenitors in their system is critical and rather than providing these data solely in revision responses and not in the main manuscript, I encourage the authors to incorporate their data into the main manuscript.

As suggested by the reviewer, we have included this figure in the revised manuscript (Supplementary Fig. 1b,c).

2. The broader issue of using CHIR and SAG (Wnt and Hedgehog agonists, respectively) as mitogens while disregarding their morphogenic effects remains confusing. These compounds are well established as having a dramatic effect on lineage commitment of neuronal progenitors and are essentially ignored by the authors without strong rationale. The rationale presented—that their use at early stages allows for primarily mitogenic activity—is noted; however, compared to other protocols, days 25–32 is not considered early within typical patterning timelines⁵. While the end point seems to be that their tissue is predominantly dorsally patterned, this significant deviation from established systems is poorly addressed. For instance, if we accept the conclusions from the immunostaining, how do the authors explain the apparent lack of morphogenic effect?

We apologize for any confusion caused by our initial response. To clarify, dorsoventral patterning of the telencephalon is established very early in human brain development (around gestational week 5; refs 1,2). Mimicking this developmental process, previous cortical organoid protocols employ CHIR99021 and SAG as patterning cues during the early specification window (days 5–20, following neuroectoderm induction) to induce dorsal versus ventral identity (refs 3–6).

In contrast, our study introduces CHIR99021 and SAG at a relatively later stage (days 25–32), following dual-SMAD inhibition (days 0–5) and an early Wnt activation phase (days 5–14), by which point the cells have already been committed to a dorsal telencephalic fate. This later stage lies well beyond the critical window for regional patterning of the telencephalon. As cortical organoids are already specified toward a dorsal lineage at this point, further activation of Hh and Wnt pathways is unlikely to act as strong morphogenic signals.

Instead, as demonstrated in our study, treatment with CHIR99021 and SAG at this stage resulted in significant expansion of the SOX2⁺ neural progenitor pool and thickening of VZ-like rosettes, while maintaining dorsal identity (Fig. 1b,c and Supplementary Fig. 1a–c), suggesting that Hh and Wnt pathways function primarily as mitogens. These findings are consistent with previous reports showing that activation of Hh and Wnt signaling promotes proliferation and expansion of neural progenitor cells in the neocortex (refs 7,8), as elaborated in our initial response.

1. Jessell, T. Neuronal specification in the spinal cord: inductive signals and transcriptional codes. *Nat Rev Genet* 1, 20–29 (2000).

2. Stiles, J., Jernigan, T.L. The Basics of Brain Development. *Neuropsychol Rev* 20, 327–348 (2010).

3. Kadoshima, T. et al. Self-organization of axial polarity, inside-out layer pattern, and species-specific progenitor dynamics in human ES cell-derived neocortex. *Proceedings of the National Academy of Sciences* 110, 20284–20289 (2013).

4. Qian, X. et al. Brain-Region-Specific Organoids Using Mini-bioreactors for Modeling ZIKV Exposure. *Cell* 165, 1238–1254 (2016).

5. Birey, F., Andersen, J., Makinson, C. et al. Assembly of functionally integrated human forebrain spheroids. *Nature* 545, 54–59 (2017).

6. Xiang, Y. et al. Fusion of Regionally Specified hPSC-Derived Organoids Models Human Brain Development and Interneuron Migration. *Cell Stem Cell* 21, 383-398 (2017).

7. Wang, L., Hou, S. & Han, Y.-G. Hedgehog signaling promotes basal progenitor expansion and the growth and folding of the neocortex. *Nature Neuroscience* 19, 888-896 (2016).

8. Neumann, J. E. et al. A mouse model for embryonal tumors with multilayered rosettes uncovers the therapeutic potential of Sonic-hedgehog inhibitors. *Nat Med* 23, 1191-1202 (2017).

3. The authors use well established markers for excitatory lineages (i.e., dorsal (PAX6 and TBR1) but only use DLX1 as validation for the lack of inhibitory neuron progenitors for which there are numerous, region specific, markers. In order to make this conclusion stronger, the authors could broaden their scope of ventral markers to include more specific identifiers of inhibitory neuron lineages, many of which have been applied already in organoids. These may include NKX2-1 (Medial Ganglionic Eminence), GSX2, LHX6 (Lateral Ganglionic Eminence), SCGN and COUP-TFII (Caudal Ganglionic Eminence), DLX2, SOX6 (migrating GABAergic progenitors), many of which have been used in other studies.

Taken together, the authors have assessed the developmental lineage of their cortical assembloids to a degree and have partially addressed earlier comments. However some broader conceptual concerns remain.

As suggested by the reviewer, we conducted additional immunostaining and quantitative analyses using a broader set of ventral progenitor markers to further assess regional identity (Fig R1; Supplementary Fig. 1b,c). Consistent with our original observations, the results confirmed that the cortical assembloids maintained a dorsal identity, with minimal or no significant expression of ventral markers detected.

Fig. R1. Immunostaining analysis of cortical assembloids to assess the expression of ventral progenitor markers.

Comment 3: The radial organization of progenitors and growing axons is impressive but needs to be supported by statistical analysis. Labelling of individual radial glia cells and individual neurons will be important to make this point. This can easily be done, for example using an adenovirus

Response and Assessment 3: The authors have addressed this point by using a combination of viral vectors expressing ubiquitous EGFP and SOX2 immunostaining. The rationale is that SOX2/EGFP co-positive cells represent radial glial populations, while SOX2-negative, EGFP-positive cells represent neurons. By quantifying the number of SOX2+ cells oriented within $\pm 30^\circ$ of a line drawn perpendicular to the tangential surface of the ventricular zone, they were able to identify radially organized cells. Overall, this approach addresses my earlier comments.

We appreciate that the reviewer is satisfied with our responses.

Comment 4: The authors use the term “assembloid” to describe their cultures. I am aware that this is in agreement with Pasca et al., Nature 2022. On the other hand, a related technology (Antón-Bolaños, et al., Nature 2024) has recently been called a “chimeroid”. I am afraid that this difference in nomenclature might be slightly confusing and would suggest the authors to add a clarifying sentence (for example, explaining why they chose assembloids over chimeroid).

Response and Assessment 4: In response to my earlier comments, the authors have provided a rationale for their choice of "assembloid" versus "chimeroid" nomenclature and therefore addressed my earlier concern.

We appreciate that the reviewer is satisfied with our responses.

Comment 5: The consistency of the MEA signals is impressive. But it might be misleading: Besides excitatory neurons, assembly of functional circuits requires inhibitory interneurons, which are not present in the cultures. In my view, this is a major weakness of the method that should be mentioned in the text.

Response and Assessment 5: The authors have partially addressed this concern by noting the absence of inhibitory interneurons in the discussion section. However, it remains unclear how this limitation is connected to the electrophysiological activity presented in Figure 3. While the authors point out that other studies recapitulating dorsal forebrain development similarly lack interneurons, they have not directly discussed how this deficiency might affect the functionality of their cortical assembloids. This point should be addressed more explicitly. Although Figure 3 demonstrates that the cortical assembloids exhibit a degree of functionality, the authors have not compared their results to other models that either include or exclude inhibitory neuron populations. This remains a significant weakness, as it is unclear whether the single-rosette method improves tissue functionality relative to existing approaches. In order to make this conclusion stronger, the authors should demonstrate, preferably experimentally, that the electrophysiological activity displayed here is on par or an improvement when compared to other similar studies which lack interneurons.

As the reviewer correctly noted, inhibitory interneurons are critical regulators of cortical network dynamics *in vivo*. They mediate both feedforward and feedback inhibition, orchestrate the synchrony of excitatory neurons, and prevent excessive firing (refs 9,10). In excitatory networks lacking inhibitory neurons, frequent and synchronized burst-firing patterns are commonly observed (refs 11,12). Consistent with these findings, our cortical assembloids exhibited frequent, robust, and highly synchronized burst activity, as measured by MEA recordings (Fig. 3f–h). This heightened neuronal activity is likely driven by increased excitatory neuron maturity, promoted by our module-based cellular reconstitution approach. We have now clarified this point in the Discussion section of the revised manuscript (page 18).

As suggested by the reviewer, we conducted comparative MEA recordings on cortical organoids generated using four widely adopted protocols, all of which similarly lack inhibitory neurons. We found that our cortical assembloids consistently exhibited stronger and highly synchronized activity compared to these existing models (Fig. R2). These results indicate that our assembloids demonstrate enhanced electrophysiological activity, likely reflecting the increased maturation of excitatory neurons achieved through our modular cellular reconstitution approach.

9. Markram, H., Toledo-Rodriguez, M., Wang, Y. et al. Interneurons of the neocortical inhibitory system. *Nat Rev Neurosci* 5, 793–807 (2004).

10. Kepecs, A., Fishell, G. Interneuron cell types are fit to function. *Nature* 505, 318–326 (2014).

11. Masquelier T, Deco G. Network Bursting Dynamics in Excitatory Cortical Neuron Cultures Results from the Combination of Different Adaptive Mechanism. PLOS ONE 8 (2013).

12. Mòdol, Laura et al. Somatostatin interneurons control the timing of developmental desynchronization in cortical networks. Neuron 112, 2015-2030 (2024).

Fig. R2. Comparative analysis of electrophysiological activity between cortical assembloids and cortical organoids reported in previous studies.

Response 6

This concern has not been fully addressed. While the author's overall conclusion that cortical assembloids display consistent properties appears valid and is an important contribution to the field, there are significant issues with how the analyses comparing their method to previously published protocols were conducted, which weakens the impact of their claims. Some major points are listed below:

1. The analysis presented in Figures 4C and 5C appears inappropriate for comparing cell type-specific and layer marker expression across protocols. The authors use different marker genes across datasets, preventing a direct and fair comparison. If the authors are using specific marker genes in their own protocol, the same markers should be applied across datasets to properly assess variability, rather than selecting different genes from different papers.

Regarding the reviewer's comment about comparing marker gene expression across datasets, we acknowledge that employing different marker genes across datasets can complicate direct comparisons. However, in single-cell RNA-seq analyses, it is technically acceptable and widely practiced to employ dataset-specific marker genes, given that absolute gene expression values can significantly vary between studies due to experimental differences such as culture protocols, duration, sequencing depth, and inherent batch effects (refs 13,14). Indeed, each of the four original studies we analyzed utilized distinct marker sets tailored specifically to best represent their respective datasets. Therefore, to faithfully reproduce and accurately represent the original findings, we used slightly different yet largely overlapping sets of widely accepted canonical marker genes for each dataset.

To further address the reviewer's concern (also suggested by reviewer 1), we performed an additional analysis by integrating our cortical assembloid data with conventional cortical organoid datasets generated using eleven independent protocols. This integrative approach, utilizing a unified set of markers, confirmed that our cortical assembloids exhibit a higher degree of consistency and maturation than previously reported cortical organoids (Supplementary Fig. 11a-

h; Supplementary Data 1), consistent with our earlier analyses of each individual dataset (Figs. 4,5 and Supplementary Figs. 6,7).

13. Luecken MD, Theis FJ. Current best practices in single-cell RNA-seq analysis: a tutorial. *Mol Syst Biol.* 19 (2019)

14. Tran HTN, et al. A benchmark of batch-effect correction methods for single-cell RNA sequencing data. *Genome Biol.* 16 (2020)

2. The authors claim that neurons in their cortical assembloids segregate into six distinct populations expressing layer-specific markers. However, this same degree of specification appears present in other datasets as well (e.g., Supplementary Figure 7), albeit in lower amounts. Therefore, the authors should limit their claims to their data, that demonstrates they can achieve robust amounts of the separate layers, but not necessarily their existence, as other studies can also achieve this.

In the manuscript, we aimed to accurately describe the characteristics of cortical assembloids, including the development of distinct populations corresponding to the six cortical layers. It was not our intention to dismiss previous studies that also reported evidence of layer specification. For instance, on page 12, we wrote: “Further in-depth analysis revealed that the neurons in cortical assembloids were segregated into six distinct cell populations expressing layer-specific markers, with each population corresponding to a specific cortical layer of the human brain.”

In response to this comment, we acknowledge that our original statement may have been somewhat misleading. Accordingly, we have revised the description of our cortical assembloids to more accurately reflect the specific advances achieved and to avoid any potential overstatement (pages 12-15).

3. The stacked bar plots in Figure 4B do not convincingly support the claim that their method significantly reduces the percentage of radial glial cells compared to other protocols as often times their cortical assembloids can consist of over 30% radial glial cells, similar to other studies. While reduced batch variability seems to be the authors' strongest argument, the assertion that other methods consistently have higher radial glial proportions than theirs is not strongly supported by the data presented.

In the original manuscript, our intention was to accurately describe the proportions of different cell types in our cortical assembloids, rather than to claim a reduction in radial glial cells compared to other existing organoid models. Specifically, on pages 12, we stated: “...cortical assembloids exhibited a high degree of consistency, containing seven transcriptionally distinct cell types within forebrain lineages, including radial glia (RG), dividing RG, intermediate progenitor cells (IPCs), excitatory neurons, a small number of inhibitory neurons, astrocytes, and microglia.”

In the revised version of our manuscript, we have further rephrased the remaining relevant sentence to provide a more accurate and balanced description of the cellular composition in the cortical assembloids (page 15).

4. The authors argue that other protocols suffer from inconsistent interneuron production. But this critique applies equally to their own system, where inhibitory neurons are also largely absent. It is therefore unclear to me how their method improves upon this.

As we acknowledge the reviewer’s comment on our description of other cortical organoids, we revised our manuscript to rephrase the portion of descriptions about inconsistent interneuron production in current cortical organoids (page 12). In addition, we clearly described limitations of

our methodology, particularly the insufficient generation of interneurons in our cortical assembloids in the Discussion section of our revised manuscript (page 18).

5. The authors note the absence of microglia in other protocols. However, it is well-established that microglia must typically be added exogenously to brain organoid cultures. This makes the comparison somewhat unfair. If the authors wish to strengthen their point regarding microglia, they should compare their findings to protocols where microglia were intentionally incorporated into organoids^{6,7} and assess survival, integration, and transcriptional fidelity (referenced in Line 288 - 289).

References

1. Quadrato, G. et al. Cell diversity and network dynamics in photosensitive human brain organoids. *Nature* 545, 48–53 (2017).
2. Uzquiano, A. et al. Proper acquisition of cell class identity in organoids allows definition of fate specification programs of the human cerebral cortex. *Cell* 185, 3770–3788.e27 (2022).
3. He, Z. et al. An integrated transcriptomic cell atlas of human neural organoids. *Nature* 635, 690–698 (2024).
4. Qian, X. et al. Sliced Human Cortical Organoids for Modeling Distinct Cortical Layer Formation. *Cell Stem Cell* 26, 766–781.e9 (2020).
5. Paşca, A. M. et al. Functional cortical neurons and astrocytes from human pluripotent stem cells in 3D culture. *Nature Methods* 12, 671–678 (2015).
6. Cakir, B., Kiral, F. R. & Park, I.-H. Advanced in vitro models: Microglia in action. *Neuron* 110, 3444–3457 (2022).
7. Popova, G. et al. Human microglia states are conserved across experimental models and regulate neural stem cell responses in chimeric organoids. *Cell Stem Cell* S1934590921003763 (2021) doi:10.1016/j.stem.2021.08.015.

We appreciate the reviewer's comment. We recognize that previous protocols have incorporated microglia exogenously through co-culture strategies (refs 15-17). However, these approaches typically fail to achieve robust integration and yield immature microglia, largely due to two main limitations: first, the uneven spatial distribution of microglia throughout the organoid, and second, the inconsistency across individual organoids in maintaining a physiologically relevant proportion of microglia relative to the total cell population. Our microinjection strategy addresses the limitations of previously reported methods by enabling precise and controlled delivery of microglia throughout the assembloids (Fig. 1j,k and Supplementary Fig. 1g). This approach facilitates the development of mature microglial populations while minimizing disruption to existing tissues (Fig. 2d and Supplementary Fig. 4a,c; Supplementary Video 2).

As suggested by the reviewer, we performed additional comparative analysis to further support the efficient integration and maturation of microglia in our cortical assembloids (Fig. R3). By carefully regulating the number of microglia injected, the timing of each injection, and the spatial distribution within the assembloid, we believe that our method offers a highly efficient and reliable strategy for integrating microglia into cortical assembloids.

15. Cakir, B., Tanaka, Y., Kiral, F.R. et al. Expression of the transcription factor PU.1 induces the generation of microglia-like cells in human cortical organoids. *Nat Commun* 13, 430 (2022).

16. Park, D.S., Kozaki, T., Tiwari, S.K. et al. iPS-cell-derived microglia promote brain organoid maturation via cholesterol transfer. *Nature* 623, 397–405 (2023).

17. Schafer, Simon T. et al. An in vivo neuroimmune organoid model to study human microglia phenotypes. *Cell* 186, 2111 – 2126 (2023).

Fig. R3. Comparative analysis of microglial integration and maturation, including spatial distribution, integration consistency, and expression of microglial-specific genes, in cortical assembloids using co-culture versus microinjection approaches. (a) Representative images of GFP+ microglia in cortical assembloids under co-culture (top) and microinjection (bottom) conditions at day 0 and day 30. Scale bars, 1 mm. (b) Quantification of microglial integration efficiency, defined as the percentage of GFP+ cells among total DAPI-stained nuclei, in three independent organoids per condition. (c) RT-qPCR analysis for the expression of various microglia markers AIF1, ITGAM, TMEM119, and CSF1.

Reviewer #3 (Remarks to the Author):

The work claims that the authors can reconstitute various cell modalities to form a highly complex, mature, homogeneous six-layered cortex, termed cortical assembloids. Showing six-layered cortices that develop from a germinal layer of neurogenesis with functionality is a significant advance, which is highly appreciated in the field. The authors use the term reconstitution, and they claimed that they could reconstitute various cellular compositions to form a structurally brain-like tissue system with six layers of cortex. Conceptually, it is a fascinating idea, even though the layered cortex does not emerge from the typical ventricular zone, and the data are of high quality.

We thank the reviewer for their appreciation of the significance, innovation, and high quality of our study, as well as for their overall satisfaction with our initial responses to their comments.

By definition, reconstitution means to break down a biochemical complex or a tissue and put it back in a functional form. This is where the concept is not solid because reconstitution is not reflected in the work. The work involves a scaffold of neural stem cells (Sox2-positive cells) and is then supplemented with reelin-positive neurons, which surprisingly and nicely guide the six layers. This is not a reconstitution, but rather driven by a sequential differentiation process, which is equally exciting. Naming it and extensively discussing the process of cellular reconstitution is misleading and incorrect. My further comments are provided below, and I hope they will be helpful to the authors in restructuring the work into an acceptable form. Data quality is, of course, of high quality, but it must be carefully analyzed to its depth to derive a solid conclusion.

Our novel methodology for developing cortical assembloids via cellular reconstitution technology builds upon our previous work (ref 1), where we developed an innovative approach to create functional tissues by three-dimensionally reconstituting tissue stem cells alongside diverse cell types from the tissue stroma. While this approach was applied to a different organ, it provided the conceptual foundation for human assembloids, defined as organoids formed by reconstituting multiple human tissue cell types. In the present study, we extended this approach to generate human cortical assembloids by reconstituting single-rosette organoids with RELN-expressing neurons and glial cells, including astrocytes and microglia, in a modular, stepwise manner. Thus, we believe the use of the term “cellular reconstitution” is acceptable and aligns with established nomenclature in the field. If the reviewer remains unsatisfied with our response, we would greatly appreciate specific suggestions and would be happy to make further modifications as needed.

1. Kim, E. et al. Creation of bladder assembloids mimicking tissue regeneration and cancer. *Nature* 588, 664-669 (2020).

The authors must avoid strong statements without any basis, which already appear in the abstract, that “Current brain organoid technology fails to provide adequate patterning cues to induce a mature structure that represents the complexity of the human brain.” Which of the ones fail to show the patterning? I could name dozens of them who have dramatically developed the methodology. I found their statement to be blunt and somewhat unfair. The presented work does not show any early patterning event (see below)

We have revised the sentence in the Abstract in our revised manuscript (page 2).

“Modular-based cellular reconstitution technology” is indeed a nice term to read, and the manuscript must explain the logic behind it. The power of the newly developed method can be witnessed only when it is used to model a disease. Otherwise, the authors’ claims remain questionable. Without presenting one or two applications, the work is merely descriptive of the current state of brain organoid technology and thus falls short of publication here.

While we fully recognize the importance of applying assembloid platforms to disease modeling, the current study is focused on the development and characterization of our novel cortical assembloid models. Given the length and scope of this manuscript, we believe that disease modeling falls beyond its intended focus. Additional studies will be required to evaluate the applicability of our cortical assembloid model for investigating the mechanistic insights of various brain disorders. For the reviewer's reference, we have published a separate study (ref 2), currently available on bioRxiv, which demonstrates the application of our cortical assembloid platform to modeling human schizophrenia.

2. Kim, et al. Identification of neuron-glia signaling feedback in human schizophrenia using patient-derived, mix-and-match forebrain assembloids. bioRxiv (2024).

The authors conducted an extensive literature review, covering over 60 papers, and presented a scoring system indicating that all fall short in some way. It seems like a gross overstatement, and I question the logic behind the scoring system. They cite reference 23, which has no relevance to brain organoids. I am unsure of its origin, other than that it is their own citation.

We appreciate the reviewer's acknowledgment of our efforts to provide a comprehensive review of brain organoid research. Regarding the reviewer's comment about our assessment criteria, we evaluated each study across four primary parameters—structural organization, functional maturity, cellular diversity, and overall heterogeneity. For each parameter, we defined multiple subcategories and conducted a comprehensive review of all available data (including text, figures, and supplementary materials) from each publication to score each subcategory. For example, in the evaluation of heterogeneity, a score of 'NA' was assigned when no experimental validation was provided for a given subcategory. Even when data were presented, we still assigned 'NA' if only a single representative image was shown. When multiple organoids were evaluated, the subcategory was scored as 'Low' if the study included representative images and quantitative analyses across multiple organoids that demonstrated clear consistency or a pronounced lack of heterogeneity among replicates; otherwise, it was scored as 'High'. Subcategory scores were then summed to obtain an overall parameter score, which was classified as high, mid, or low based on predefined thresholds. The detailed scoring criteria are provided in Supplementary Data 2 of the original manuscript.

We acknowledge that, despite our systematic evaluation, some subjectivity may remain. In response to the reviewer's comments, we therefore decided to retain these comprehensive reviews (Supplementary Table 1; Supplementary Data 1 and 2) solely as reference materials for the reviewers during revision and have removed them from the main text. We would be, of course, happy to reincorporate it into the manuscript if the reviewer believes it would be beneficial.

Regarding reference 23, we cited this work to provide the conceptual foundation for the cellular reconstitution strategy used to generate human assembloids. As noted above, the novel technology presented in this manuscript builds upon our previous work (ref 1), in which we developed an innovative approach to create functional tissues by three-dimensionally reconstituting tissue stem cells alongside diverse stromal cell types. Although that study focused on a different organ system, it established the foundational concept of generating human assembloids—defined here as organoids created by reconstituting multiple cell types derived from human tissues. While we believe that citing this work in the current context is appropriate, we would, of course, be happy to remove the citation if the reviewer has any remaining concerns.

The modular assembly shown in their work is very challenging and complex, and I appreciate it as the authors have tried it with at least three iPS donors and an ES cell line. The scheme illustrates the general method, but the data lacks details. How were the organoids dissected manually to get one rosette containing tissue, and how is it possible to do this in large numbers without any variations? The method also does not control the number of pluripotent cells used (says 2-3 clones or something). SAG and CHIR (agonists of growth signalling) opposes each other. For example, elevated SHH signaling in vivo prevents the neural tube from closing, which explains why the authors observe a large rosette-like structure. Then, what was the logic in using CHIR in addition to SAG? How could the authors count SOX2-positive cells in a rosette of a 3D tissue?

First, detailed information on the single-rosette dissection is already provided in both the Results and Methods sections of our manuscript (Fig. 1a; page 23). For clarification, approximately 5-7 single-rosettes were formed in one conventional brain organoid. Given that each batch of PSCs typically produces about 5-8 conventional organoids, this results in around 25-56 single-rosette organoids per batch. Our analysis shows that the sizes of these single-rosette organoids derived from multiple hPSCs follow a normal distribution, making the median size a meaningful representative measure. To ensure uniformity, we selected five single-rosette organoids with diameters closest to the median, thereby reducing variation across organoids and batches while preserving the unbiased biology of individual conventional organoids.

Second, it appears that the reviewer is questioning about the role of CHIR99021 and SAG. To clarify, CHIR99021 and SAG, which activate Wnt and Hedgehog (Hh) signaling, respectively, are employed here not to modulate regional identity but to serve as potent mitogens. While we acknowledge the critical roles of Hh and Wnt as major morphogens in the development of the human embryo, dorsoventral patterning of the telencephalon is established very early in human brain development (gestational week 5; refs 3,4). Mimicking this developmental process, previous cortical organoid protocols employ CHIR99021 and SAG as patterning cues during the early specification window (e.g., days 5–10 post-neuroectoderm induction) to induce dorsal versus ventral identity (refs 5-8). In contrast, our study introduces CHIR99021 and SAG at a relatively later stage (days 25–32), following dual-SMAD inhibition (days 0–5) and an early Wnt activation phase (days 5–14), by which point the cells have already been committed to a dorsal telencephalic fate. This later stage lies well beyond the critical window for regional patterning of the telencephalon. As cortical organoids are already specified toward a dorsal lineage at this point, further activation of Hh and Wnt pathways is unlikely to act as strong morphogenic signals. Instead, as demonstrated in our study, treatment with CHIR99021 and SAG at this stage resulted in significant expansion of the SOX2⁺ neural progenitor pool and thickening of VZ-like rosettes, while maintaining dorsal identity (Fig. 1b,c and Supplementary Fig. 1a–c), suggesting that Hh and Wnt pathways function primarily as mitogens. These findings are consistent with previous reports showing that activation of Hh and Wnt signaling promotes proliferation and expansion of neural progenitor cells in the neocortex (refs 9,10).

Third, regarding the quantification of SOX2-positive cells, we counted the number of SOX2-positive nuclei within each rosette across two representative sections per organoid. The detailed quantification methods are provided in the figure legend (Fig. 1c).

3. Schuurmans C, Guillemot F. Molecular mechanisms underlying cell fate specification in the developing telencephalon. *Curr Opin Neurobiol.* 12, 26-34 (2002).

4. Stiles, J., Jernigan, T.L. The Basics of Brain Development. *Neuropsychol Rev* 20, 327–348 (2010).

5. Kadoshima, T. et al. Self-organization of axial polarity, inside-out layer pattern, and species-specific progenitor dynamics in human ES cell-derived neocortex. *Proceedings of the National Academy of Sciences* 110, 20284-20289 (2013).
6. Qian, X. et al. Brain-Region-Specific Organoids Using Mini-bioreactors for Modeling ZIKV Exposure. *Cell* 165, 1238-1254 (2016).
7. Birey, F., Andersen, J., Makinson, C. et al. Assembly of functionally integrated human forebrain spheroids. *Nature* 545, 54–59 (2017).
8. Xiang, Y. et al. Fusion of Regionally Specified hPSC-Derived Organoids Models Human Brain Development and Interneuron Migration. *Cell Stem Cell* 21, 383-398 (2017).
9. Wang, L., Hou, S. & Han, Y.-G. Hedgehog signaling promotes basal progenitor expansion and the growth and folding of the neocortex. *Nature Neuroscience* 19, 888-896 (2016).
10. Neumann, J. E. et al. A mouse model for embryonal tumors with multilayered rosettes uncovers the therapeutic potential of Sonic-hedgehog inhibitors. *Nat Med* 23, 1191-1202 (2017).

I don't question the technical soundness of the method, but it lacks content and details. It is, by far, a very complex technology to adapt to reproduce and simultaneously minimize intra- and inter-batch differences. With this complexity, it is challenging to generate a large number of organoids, and it is unarguably understandable.

In response to this comment, we have revised the text in the Discussion section of our manuscript to describe the limitations of our methodology, particularly the limited scalability of cortical assembloid generation (page 19).

Other fundamental problems here are the apicobasal polarity, which must be demonstrated. I don't see any indication for neurons in Shh and CHIR-treated organoids (before they seed reelin cells), and they are just too big and don't look like there is a clear apicobasal polarity. In fact, for the first few days, a method to generate forebrain organoids has been adapted. How come there are no neurons at the basal side? From this, I seriously suspect that the Sox2-positive structures are merely unphysiological, and this must be tested for the apicobasal markers, such as ZO1 and the pan-neuronal marker TUJ1. I guess that the shown are a cluster of SOX2-positive cells in a perturbed, enlarged, and broken neural rosette. In the displayed images, I do see two or three rosette-like structures of two different sizes (in contrast to what the authors claim). Again, the authors have provided other cell types, which were cultured separately, a notable achievement, but one that may not be directly related to embryonic brain developmental cues. This is already obvious as their organoid does not generate inhibitory neurons. A control experiment should be shown in which the Sox2-positive rosette develops without supplementing reelin cells.

The reviewer appears to question whether apicobasal polarity is maintained in our early-stage cortical organoids following treatment with CHIR99021 and SAG. To address this concern, we performed immunostaining for SOX2, TUJ1, and ZO-1 on cortical organoids after treatment. We observed that these organoids preserved proper apicobasal polarity, as indicated by ZO-1 localization at the apical luminal surface, and TUJ1+ neurons along the basal surface of the SOX2+ thickened, VZ-like rosettes (Fig. R4). These enlarged, single rosette structures further maintained apicobasal polarity following single rosette dissection (Fig. 1d).

Fig. R4. Immunostaining analysis of early cortical organoids treated with CHIR99021 and SAG for SOX2, TUJ1, and ZO-1.

Regarding the reviewer’s comment on “brain developmental cues,” it appears to stem from a misunderstanding of our methodology for generating cortical assembloids. To clarify this point, we have provided additional explanation below.

First, Reelin is a well-established developmental signal, expressed by the earliest cortical layer neurons migrating into the marginal zone during early human corticogenesis, and is required for proper neural guidance and laminar organization (refs 11,12). We therefore reconstituted Reelin-expressing cells to the basal surface of the single rosette to drive the establishment of six distinct cortical layers. Second, astrocytes and microglia were introduced to enhance the cellular diversity of the cortical organoids, not to serve as developmental cues for cortical layer formation. These cells were integrated into the cortical assembloids at a later stage, after the initial structuring of the cortical layers had occurred.

To address the specific question regarding the experiments without RELN+ cells, we already conducted additional experiments to examine the structural maturity of cortical assembloids in the absence of RELN+ cells during the first round of revisions. Our experimental data demonstrated that, in the absence of RELN+ cells, the assembloids developed disorganized cortical layers with a marked reduction in upper-layer neurons (Fig. R5), indicating that RELN+ cells are primarily responsible for proper cortical layer formation in our cortical assembloids.

Fig. R5. Immunostaining analysis of the six cortical layers in cortical assembloids lacking a RELN+ layer.

11. Causeret, F., Moreau, M. X., Pierani, A. & Blanquie, O. The multiple facets of Cajal-Retzius neurons. *Development* 148 (2021).

12. Bystron, I., Blakemore, C. & Rakic, P. Development of the human cerebral cortex: Boulder Committee revisited. *Nature Reviews Neuroscience* 9, 110-122 (2008).

Six layers of neurons are the most exciting part of the work. I am curious how the authors could combine 6 markers in one section. Or are these adjacent sections? Then, how do they create a nice layer over 24 μm? It would be critical to show more than one specific markers for each layer.

As detailed in the figure legends, we used three adjacent 8-μm sections, each immunostained for TBR1/CTIP2, SATB2/RELN, or BRN2/CUX2, respectively, to cover all six cortical markers across a 24-μm span.

Connect figure 2a to figure 1. The rest of the analysis is basically a consequence of those 6-layered neurons, which have been well performed.

As suggested by the Reviewer, we have repositioned Fig. 2a as panel I in Fig. 1.

Sc-RNA sequencing has been performed and mainly used to compare the cell proportions, which is not very well exploited (again, figures lack units, and as a result, I don't know what those bars are. Secondly, unsupervised clustering of cell types was not shown). In normal brain development, the excitatory neurons emerge from the dorsal telencephalon, and the inhibitory neurons are from the ventral telencephalon (in this work, inhibitory neurons are low). The data shows some sort of trajectory analysis, but the text does not provide the details. Essentially, the authors describe it as the "trajectory follows normal brain development," but what constitutes normal here?

Regarding the reviewer's request to provide the units, we have added y-axis labels to the bar graphs to specify the units and clearly indicate what each bar represents (Fig. 4b and Fig. 5b).

To address the question about developmental trajectory, we have revised the text related to the trajectory analysis to provide a more precise description of our data (page 12). The excitatory neurons within each cortical layer of our assembloids follow the developmental sequence, generating layer-specific neurons in order from layer 6 to layer 2 (Fig. 5d), closely mirroring the temporal pattern observed in the developing human cortex (ref 13). As the reviewer noted, this analysis is limited to excitatory neurons due to the relatively low proportion of inhibitory neurons in our assembloids. As elaborated in our initial response, our method utilizes dorsally patterned cortical organoids, which predominantly generate excitatory neurons while producing only a small population of interneurons. This characteristic is consistent with many existing cortical organoid models that reflect the intrinsic developmental programs of the dorsal forebrain, where inhibitory interneurons are not readily produced (refs 14–16). We have described this limitation—specifically, the insufficient generation of interneurons—in the Discussion section of the revised manuscript (page 18).

13. Cadwell, C. R., Bhaduri, A., Mostajo-Radji, M. A., Keefe, M. G. & Nowakowski, T. J. Development and Arealization of the Cerebral Cortex. *Neuron* 103, 980-1004 (2019).

14. Paşca, A., Sloan, S., Clarke, L. et al. Functional cortical neurons and astrocytes from human pluripotent stem cells in 3D culture. *Nat Methods* 12, 671–678 (2015).

15. Yoon, S.J., Elahi, L.S., Paşca, A.M. et al. Reliability of human cortical organoid generation. *Nat Methods* 16, 75–78 (2019).

16. Qian, X. et al. Sliced Human Cortical Organoids for Modeling Distinct Cortical Layer Formation. *Cell Stem Cell* 26, 766-781.e769 (2020).

There are several issues, and one such issue is injecting microglia in 5 different places, and this is quite an invasive method and will induce injuries to the organoids. Why not spike microglia and test if they are taken up and integrated? The authors should consider softening the discussion section when comparing their work with other published results that have modeled one or more diseases, demonstrating that the tissues are functional. The current work lacks any application.

Regarding the comments about microglia, it is important to note that previous protocols have incorporated microglia exogenously through co-culture strategies (refs 17-19). However, these approaches typically yield immature microglia and fail to achieve robust integration, largely due to two main limitations: first, the uneven spatial distribution of microglia throughout the organoid, and second, the inconsistency across individual organoids in maintaining a physiologically relevant proportion of microglia relative to the total cell population. Our microinjection strategy addresses the limitations of previously reported methods by enabling precise and controlled delivery of microglia throughout the assembloids (Fig. 1j,k and Supplementary Fig. 1g). This approach facilitates the development of mature microglial populations while minimizing disruption to existing tissues (Fig. 2d and Supplementary Fig. 4a,c; Supplementary Video 2).

In addition, we performed additional experiments to further support the efficient integration and maturation of microglia in our cortical assembloids, as shown in Fig. R3 in our response to Reviewer 2. By carefully regulating the number of microglia injected, the timing of each injection, and the spatial distribution within the assembloid, we believe that our method offers a highly efficient and reliable strategy for integrating microglia into cortical assembloids.

17. Cakir, B., Tanaka, Y., Kiral, F.R. et al. Expression of the transcription factor PU.1 induces the generation of microglia-like cells in human cortical organoids. *Nat Commun* 13, 430 (2022).

18. Park, D.S., Kozaki, T., Tiwari, S.K. et al. iPS-cell-derived microglia promote brain organoid maturation via cholesterol transfer. *Nature* 623, 397–405 (2023).

19. Schafer, Simon T. et al. An in vivo neuroimmune organoid model to study human microglia phenotypes. *Cell* 186, 2111 – 2126 (2023).

Regarding the comments about disease modeling using our cortical assembloids, we fully recognize the importance of applying assembloid platforms to disease modeling. However, as elaborated above, the current study is focused on the development and characterization of our novel cortical assembloid models. Given the length and scope of this manuscript, we believe that disease modeling falls beyond its intended focus. Additional studies will be required to evaluate the applicability of our cortical assembloid model for investigating the mechanistic insights of various brain disorders. For the reviewer's reference, we have published a separate study (ref 2), currently available on bioRxiv, which demonstrates the application of our cortical assembloid platform to modeling human schizophrenia. Regarding the revision of our statements in the Discussion section, we would greatly appreciate specific suggestions and would be happy to make further modifications accordingly.

Overall, starting with the Sox2 scaffold, the addition of reelin cells forming a sphere with functional neurons is interesting work that requires restructuring, careful analysis, and avoiding overstating statements will develop valuable work.

We appreciate the reviewer's valuable and constructive comments.